# Tight Lower Bounds under Asymmetric High-Order Hölder Smoothness and Uniform Convexity

**Cedar Site Bai**
Department of Computer Science
Purdue University
West Lafayette, IN, USA
`bai123@purdue.edu`

**Brian Bullins**
Department of Computer Science
Purdue University
West Lafayette, IN, USA
`bbullins@purdue.edu`

## Abstract

In this paper, we provide tight lower bounds for the oracle complexity of minimizing high-order Hölder smooth and uniformly convex functions. Specifically, for a function whose $p^{th}$-order derivatives are Hölder continuous with degree $\nu$ and parameter $H$, and that is uniformly convex with degree $q$ and parameter $\sigma$, we focus on two asymmetric cases: (1) $q > p + \nu$, and (2) $q < p + \nu$. Given up to $p^{th}$-order oracle access, we establish worst-case oracle complexities of $\Omega\left(\left(\frac{H}{\sigma}\right)^{\frac{2}{3(p+\nu)-2}}\left(\frac{\sigma}{\epsilon}\right)^{\frac{2(q-p-\nu)}{q(3(p+\nu)-2)}}\right)$ in the first case with an $\ell_\infty$-ball-truncated-Gaussian smoothed hard function and $\Omega\left(\left(\frac{H}{\sigma}\right)^{\frac{2}{3(p+\nu)-2}} + \log\log\left(\left(\frac{\sigma^{p+\nu}}{H^q}\right)^{\frac{1}{p+\nu-q}}\frac{1}{\epsilon}\right)\right)$ in the second case, for reaching an $\epsilon$-approximate solution in terms of the optimality gap. Our analysis generalizes previous lower bounds for functions under first- and second-order smoothness as well as those for uniformly convex functions, and furthermore our results match the corresponding upper bounds in this general setting.

## 1 Introduction

With the advancement in computational power, high-order optimization methods ($p^{th}$-order with $p \geq 2$) are gaining more attention for their merit of faster convergence and higher precision. Consequently, uniformly convex problems (with degree $q$) have become a recent focus, particularly the subproblems of some high-order optimization methods. The subproblem of the cubic-regularized Newton ($p = 2, q = 3$) (Nesterov & Polyak, 2006) is an example, as are methods of even higher orders ($p \geq 3$, $q \geq 4$) (Zhu & Cartis, 2022).

Although these problems are high-order smooth by definition, a lower-order algorithm may be employed to obtain an approximate solution. For instance, solving the subproblem of cubic-regularized (i.e., $q = 3$) Newton with gradient descent (accessing first-order oracle, i.e., $p = 1$), or, more generally, approximately solving the subproblem of $(q - 1)^{th}$-order Taylor descent (Bubeck et al., 2019) (which typically contains a regularization term to the power of $q$) with lower-order oracle access, introduces an asymmetry between the algorithm's oracle access order and the degree of uniform convexity ($q > p + 1$).

Conversely, a lower-degree regularization can be paired with a higher-order smooth function. This enables methods that access higher-order oracles, which leads to the opposite asymmetry ($q < p + 1$). Examples include the objective function of logistic regression, which is known to be infinite-order smooth. Coupled with standard $\ell_2$-regularization, the problem can be analyzed as a $p^{th}$-order smooth and strongly convex ($q = 2$) problem, e.g., $p = 2$ with access to the Hessian matrix, $p = 3$ accessing the third-order derivative tensor.

In addressing specific instances of this asymmetry, previous works established some upper bounds (Gasnikov et al., 2019; Song et al., 2021) and lower bounds (Arjevani et al., 2019; Kornowski

& Shamir, 2020; Doikov, 2022; Thomsen & Doikov, 2024) for the oracle complexity. Notably, Song et al. (2021) proposed a unified acceleration framework for functions that are $p^{th}$-order Hölder smooth with degree $\nu$, and uniformly convex with degree $q$, providing upper bounds for any combination of $p$, $q$, and $\nu$. For the case where $q > p + \nu$, they show an oracle complexity of $\mathcal{O}\left(\left(\frac{H}{\sigma}\right)^{\frac{2}{3(p+\nu)-2}}\left(\frac{\sigma}{\epsilon}\right)^{\frac{2(q-p-\nu)}{q(3(p+\nu)-2)}}\right)$, and for the case where $q < p + \nu$, the complexity is $\mathcal{O}\left(\left(\frac{H}{\sigma}\right)^{\frac{2}{3(p+\nu)-2}} + \log\log\left(\left(\frac{\sigma^{p+\nu}}{H^q}\right)^{\frac{1}{p+\nu-q}}\frac{1}{\epsilon}\right)\right)$. To the best of our knowledge, no lower bounds exist in this general setting, particularly with Hölder smoothness and uniform convexity.

In this paper, we provide matching lower bounds to the upper bounds in (Song et al., 2021) for these asymmetric cases. Specifically, we establish $\Omega\left(\left(\frac{H}{\sigma}\right)^{\frac{2}{3(p+\nu)-2}}\left(\frac{\sigma}{\epsilon}\right)^{\frac{2(q-p-\nu)}{q(3(p+\nu)-2)}}\right)$ for $q > p + \nu$ and $\Omega\left(\left(\frac{H}{\sigma}\right)^{\frac{2}{3(p+\nu)-2}} + \log\log\left(\left(\frac{\sigma^{p+\nu}}{H^q}\right)^{\frac{1}{p+\nu-q}}\frac{1}{\epsilon}\right)\right)$ for $q < p + \nu$. For the $q > p + \nu$ case, we adopt the framework proposed by (Guzmán & Nemirovski, 2015), utilizing a smoothing operator to generate a high-order smooth function. We propose the use of $\ell_\infty$-ball-truncated Gaussian smoothing, which, as we later justify, is novelly designed to achieve the optimal rate and be compatible with both high-order smooth and uniformly convex settings. Both the truncated Gaussian smoothing and the construction of the $\ell_\infty$ ball are crucial to improve upon the sub-optimal derivation using uniform smoothing within an $\ell_2$ ball in (Agarwal & Hazan, 2018). Our results generalize the lower bounds in (Doikov, 2022; Thomsen & Doikov, 2024) to higher-order and Hölder smooth settings. For the $q < p + \nu$ case, we adopt Nesterov's framework (Nesterov et al., 2018) and generalize the lower bounds in (Arjevani et al., 2019; Kornowski & Shamir, 2020) to include Hölder smooth and uniformly convex settings.

## 2 RELATED WORK

**Upper Bounds.** Doikov & Nesterov (2021) showcase the upper bound for uniformly convex functions with Hölder-continuous Hessian via cubic regularized Newton method, but the rate is not optimal. For higher order result, Bubeck et al. (2019) and Jiang et al. (2019) established a near optimal upper bound of $\tilde{\mathcal{O}}\left(\epsilon^{-\frac{2}{3p+1}}\right)$ in the simpler case of $\nu = 1$ without uniform convexity. Gasnikov et al. (2019) achieve the same near-optimal rate, but also consider uniform convexity, and by the restarting mechanism, derive the rate that for $q > p + 1$ as well, generalizing the upper bounds established in second-order (Monteiro & Svaiter, 2013) and matching the lower bounds later derived in (Kornowski & Shamir, 2020). Kovalev & Gasnikov (2022) closed the $\log\left(\frac{1}{\epsilon}\right)$ gap, but does not consider uniform convexity or Hölder smoothness. For minimizing uniformly convex functions, Juditsky & Nesterov (2014) and Roulet & d'Aspremont (2017) study the complexity of first-order methods. Recently, Song et al. (2021) establish the most general upper bounds for arbitrary combinations of the order of Hölder smoothness and the degree of uniform convexity, which include the rates for both $q > p + \nu$ and $q < p + \nu$ cases.

**Lower Bounds.** Agarwal & Hazan (2018) proved for $p^{th}$-order smooth convex functions an $\Omega\left(\epsilon^{-\frac{2}{5p+1}}\right)$ lower bound based on constructing the hard function with randomized smoothing uniformly over a unit ball. But their rate is not optimal due to the extra dimension factor appearing in the smoothness constant due to the uniform randomized smoothing. Garg et al. (2021) added softmax smoothing prior to randomized smoothing, achieving a near-optimal rate of $\Omega\left(\epsilon^{-\frac{2}{3p+1}}\right)$ for randomized and quantum algorithms. Separately, Arjevani et al. (2019) also established the optimal lower bound of $\Omega\left(\epsilon^{-\frac{2}{3p+1}}\right)$ with the Nesterov's hard function construction approach. Furthermore, for the asymmetric case of $q < p + 1$, Arjevani et al. (2019) proved the lower bound of $\Omega\left(\left(\frac{H}{\sigma}\right)^{\frac{2}{7}} + \log\log\left(\frac{\sigma^3}{H^2}\epsilon^{-1}\right)\right)$ for the $p = 2$ and $q = 2$ case, and the result is later generalized to the $p^{th}$ order in (Kornowski & Shamir, 2020). No $q > 2$ uniformly convex settings were considered in these works. For the case of $q > p + \nu$, lower bounds for uniformly convex functions for $q \geq 3$ are limited to the first-order smoothness setting where $p = 1$ (Juditsky & Nesterov, 2014; Doikov,

2022; Thomsen & Doikov, 2024). No lower bounds for uniformly convex functions were established, to our knowledge, in the high-order setting.

# 3 PRELIMINARIES AND SETTINGS

**Notations.** We use $[n]$ to represent the set $\{1, 2, ..., n\}$. We use $\|\cdot\|$ to denote an $\ell_2$ operator norm. We use $\nabla$ for gradients, $\partial$ for subgradients, and $\langle \cdot, \cdot \rangle$ for inner products. Related to the algorithm, bold lower letters for vectors (e.g., $\mathbf{x}$, $\mathbf{y}$), and with subscript, the vectors in different iterations (e.g., $\mathbf{x}_T$). We use regular lower letters for scalars, and with subscript, a coordinate of a vector (e.g., $x_i$). Depending on the context, we use capital letters for a matrix or a random variable. We use $\phi$ for the probability density function of the standard normal or the standard multivariate normal (MVN), and $\Phi$ for the cumulative (density) function of standard normal or MVN. We further overuse the notation of $\phi_{[\cdot,\cdot]}$ $\Phi_{[\cdot,\cdot]}$ for their truncated counterparts for the normal distribution (standard normal if not specified with parameters), and $\phi_{\|\cdot\|_\infty \leq \cdot}$ $\Phi_{\|\cdot\|_\infty \leq \cdot}$ for the MVN truncated within an $\ell_\infty$ ball.

## 3.1 DEFINITIONS

**Definition 1** (High-order Smoothness). *For $p \in \mathbb{Z}^+$, a function $f : \mathbb{R}^d \to \mathbb{R}$ is $p^{th}$-order smooth or whose $p^{th}$- derivatives are $L_p$-Lipschitz if for $L_p > 0$, $\forall\, \mathbf{x}, \mathbf{y} \in \mathbb{R}^d$, $\|\nabla^p f(\mathbf{x}) - \nabla^p f(\mathbf{y})\| \leq L_p \|\mathbf{x} - \mathbf{y}\|$.*

**Definition 2** (High-order Hölder Smoothness). *For $p \in \mathbb{Z}^+$, a function $f : \mathbb{R}^d \to \mathbb{R}$ is $p^{th}$-order Hölder smooth or has Hölder continuous $p^{th}$-order derivatives if for $\nu \in (0, 1]$ and $H > 0$, $\forall\, \mathbf{x}, \mathbf{y} \in \mathbb{R}^d$, $\|\nabla^p f(\mathbf{x}) - \nabla^p f(\mathbf{y})\| \leq H \|\mathbf{x} - \mathbf{y}\|^\nu$.*

**Definition 3.** (Uniform Convexity (Nesterov et al., 2018, Section 4.2.2)) *For integer $q \geq 2$ and $\sigma > 0$, a function $f : \mathbb{R}^d \to \mathbb{R}$ is uniformly convex with degree $q$ and modulus $\sigma$ if $\forall\, \mathbf{x}, \mathbf{y} \in \mathbb{R}^d$, $f(\mathbf{y}) - f(\mathbf{x}) - \langle \nabla f(\mathbf{x}), \mathbf{y} - \mathbf{x} \rangle \geq \frac{\sigma}{q} \|\mathbf{y} - \mathbf{x}\|^q$, or the function satisfies $\langle \nabla f(\mathbf{y}) - \nabla f(\mathbf{x}), \mathbf{y} - \mathbf{x} \rangle \geq \sigma \|\mathbf{y} - \mathbf{x}\|^q$.*

# 4 LOWER BOUND FOR THE $q > p + \nu$ CASE

The derivation of the lower bound is to find such a function by construction that satisfies the uniformly convex and Hölder smooth conditions and requires at least a certain amount of iterations to reach an $\epsilon$-approximate solution. The general steps follow from the framework of showing lower complexity bounds for smooth convex optimization (Guzmán & Nemirovski, 2015), which originates from (Nemirovskii & Nesterov, 1985) and serves as the basis for results in various follow-up settings (Agarwal & Hazan, 2018; Garg et al., 2021; Doikov, 2022). The construction starts from a non-smooth function, then smooths the function with some smoothing operator (e.g. Moreau envelope in (Guzmán & Nemirovski, 2015; Doikov, 2022), randomized smoothing uniformly within a ball in (Agarwal & Hazan, 2018; Garg et al., 2021)). We design a truncated Gaussian smoothing operator within the $\ell_\infty$ ball and start the derivation by stating its formal definition and key properties.

## 4.1 TRUNCATED GAUSSIAN SMOOTHING

**Definition 4** (Truncated Gaussian Smoothing). *For $f : \mathbb{R}^d \to \mathbb{R}$ and a parameter $\rho > 0$, define the truncated Gaussian smoothing operator $S_\rho[f] : (\mathbb{R}^d \to \mathbb{R}) \to (\mathbb{R}^d \to \mathbb{R})$ as*

$$S_\rho[f](\mathbf{x}) = \mathbb{E}_V [f(\mathbf{x} + \rho V)]$$

*where $V$ is a $d$-dimensional random variable that follows the standard multivariate normal (MVN) distribution truncated within a unit ball. That is, the probability density function (PDF) of $V$ is*

$$\mathbb{P}[V = \mathbf{v}] = \frac{1}{Z(d)(2\pi)^{\frac{d}{2}}} \exp\left\{ -\frac{\mathbf{v}^\top \mathbf{v}}{2} \right\} \mathbb{I}_{[\|\mathbf{v}\|_\infty \leq 1]},$$

*in which $\mathbb{I}_{[\cdot]} = 1$ if $\cdot$ is true $0$ otherwise is the indicator function and $Z(d)$ is the normalizing factor, i.e., the cumulative distribution within the $d$-dimensional unit $\ell_\infty$-ball (Cartinhour, 1990).*

*We denote $f_\rho = S_\rho[f]$, and use the shorthand notation for the function that applied the smoothing operator for $p$ times: $f_\rho^p = S_\rho^p[f] = S_\rho[\cdots [S_\rho[f]] \cdots]$ for $p$ times.*

Now we justify the choice of truncated Gaussian smoothing for the construction of hard function. We notice that Agarwal & Hazan (2018) choose randomized smoothing uniformly over a unit $\ell_2$-ball, which by their Lemma 2.3 that the smoothed function is $\mathcal{O}(d)$-smooth (which in fact can be tightened to $\mathcal{O}(\sqrt{d})$ by (Yousefian et al., 2012; Duchi et al., 2012, Lemma 8)) where $d$ is the dimension of the variable. Since the number of iteration $T \in \mathcal{O}(d)$, their result $\mathcal{O}\left(T^{-\frac{2}{5p+1}}\right)$ is sub-optimal by an extra $T$ comparing to the tight lower bound $\mathcal{O}\left(T^{-\frac{2}{3p+1}}\right)$ (Arjevani et al., 2019). Therefore we search for a smoothing operator with Lipschitz constant being *dimension-free*. We notice that Gaussian smoothing (Duchi et al., 2012, Lemma 9), softmax smoothing (Bullins, 2020, Lemma 7), and Moreau smoothing (Doikov, 2022, Lemma 1) are such operators.

Yet as the reader will later see in the proof that the converging points are generated through a sequence of functions, instead of those generated from one hard function. For these two sequences of points to be identical so that the lower bound is indeed for optimizing the hard function constructed, we need the smoothing operator to be *local*, that is, accessing information within *some neighborhood* of the queried point, e.g., a unit $\ell_2$-ball in (Doikov, 2022). Unfortunately, Gaussian smoothing and softmax smoothing need access to global information.

For Moreau smoothing that indeed depends on local information, it's successfully applied in proving the lower bound in the first-order setting (Doikov, 2022), but is not suited for the high-order setting. First, one may attempt the extension of Moreau smoothing with a $p^{th}$-power regularization, yet it can be shown that the function is not $p^{th}$-order smooth. Next, one may try to apply Moreau smoothing $p$ times, yet unlike randomized smoothing in (Agarwal & Hazan, 2018), the Lipschitz constant does not raise to the $p^{th}$-power with the number of times the smoothing operator is applied, which leads to the same rate as in the first order. Observing the proof of (Agarwal & Hazan, 2018, Corollary 2.4), this is in essence due to the fact that the minimization in Moreau smoothing does not commute with derivative, whereas the expectation in randomized smoothing does.

We then come up with the idea of a truncated multivariate Gaussian smoothing operator that is (i) local (ii) smooth with a dimension-free constant (iii) $p^{th}$-order smooth with smoothness constant raising to the $p^{th}$ power as well. Initially, we applied the Gaussian smoothing truncated within a unit ball in $\ell_2$ by default. We noticed later, however, that the marginal distribution of unit-$\ell_2$-ball truncated multivariate Gaussian is not the truncated standard normal between $[-1, 1]$, but with an extra $d$-dependent normalizing constant, which adds the $d$-dependency to the smoothness constant of the hard function.

To ensure a dimension-free smoothness constant, we instead apply the multivariate Gaussian smoothing truncated within an $\ell_\infty$ ball, a.k.a., the hypercube with edge length 2, whose marginal distribution is indeed the truncated standard normal between $[-1, 1]$ (Cartinhour, 1990). The following lemma characterizes these desired properties including convexity, continuity, approximation, and smoothness, with proof deferred to Appendix A.1.

**Lemma 1.** *Given a L-Lipschitz function $f$, the function $f_\rho^p = S_\rho[\cdots[S_\rho[f]]\cdots]$ satisfies*

*(i) If $f$ is convex, $f_\rho^p$ is convex and L-Lipschitz with respect to the $\ell_2$ norm.*

*(ii) If $f$ is convex, $f(\mathbf{x}) \leq f_\rho^p(\mathbf{x}) \leq f(\mathbf{x}) + \frac{5p}{4} L\rho\sqrt{d}$.*

*(iii) $\forall i \in [p], \forall \mathbf{x}, \mathbf{x}' \in \mathbb{R}^d, \|\nabla^i f_\rho^p(\mathbf{x}) - \nabla^i f_\rho^p(\mathbf{x}')\| \leq \left(\frac{2}{\rho}\right)^i L\|\mathbf{x} - \mathbf{x}'\|.$*

## 4.2 THE LOWER BOUND: FUNCTION CONSTRUCTION AND TRAJECTORY GENERATION

**Theorem 1.** *For any $T$-step $(\sqrt{d} - 1 \leq T \leq d)$ deterministic algorithm $\mathcal{A}$ with oracle access up to the $p^{th}$ order, there exists a convex function $f(\mathbf{x})$ whose $p^{th}$-order derivative is Hölder continuous of degree $\nu$ with modulus $H$ and a corresponding $F(\mathbf{x}) = f(\mathbf{x}) + \frac{\sigma}{q}\|\mathbf{x}\|^q$ with regularization that is uniformly convex of degree $q$ with modulus $\sigma$, such that $q > p + \nu$, it takes*

$$T \in \Omega\left(\left(\frac{H}{\sigma}\right)^{\frac{2}{3(p+\nu)-2}} \left(\frac{\sigma}{\epsilon}\right)^{\frac{2(q-p-\nu)}{q(3(p+\nu)-2)}}\right)$$

*steps to reach an $\epsilon$-approximate solution $\mathbf{x}_T$ satisfying $F(\mathbf{x}_T) - F(\mathbf{x}^*) \leq \epsilon$.*

*Proof.* We begin the proof by constructing the hard function.

### 4.2.1 Function Construction with Truncated Gaussian Smoothing

*1. Non-smooth Function Construction.* We first construct the function

$$g_t(\mathbf{x}) = \max_{1 \le k \le t} r_k(\mathbf{x}) \qquad where \qquad \forall\, k \in [T], r_k(\mathbf{x}) = \xi_k \left\langle \mathbf{e}_{\alpha(k)}, \mathbf{x} \right\rangle - (k-1)\delta.$$

$\xi_k \in \{-1, 1\}$, $\mathbf{e}$ is the standard basis, $\alpha$ is a permutation of $[T]$, and $\delta > 0$ is some parameter that we will choose later. Lemma 2 characterizes the properties of $g_t$ with proof in Appendix A.2.

**Lemma 2.** $\forall\, t \in [T]$, $g_t$ is convex and 1-Lipschitz with respect to the $\ell_\infty$-norm, and also the $\ell_2$-norm.

*2. Truncate Gaussian Smoothing.* Next, we smooth the function $g_t(\mathbf{x})$ with truncate Gaussian smoothing as in Definition 4. Given a parameter $\rho > 0$ and $p \in \mathbb{Z}^+$,

$$G_t(\mathbf{x}) = S_\rho^p[g_t](\mathbf{x})$$

Based on Lemma 1, we show that $G_t(\mathbf{x})$ satisfies the following lemma, with proof in Appendix A.2.

**Lemma 3.** $\forall\, t \in [T]$, $\forall\, \mathbf{x}, \mathbf{y} \in \mathbb{R}^d$,

(i) $G_t(\mathbf{x})$ is convex and 1-Lipschitz, i.e., $G_t(\mathbf{x}) - G_t(\mathbf{y}) \le \|\mathbf{x} - \mathbf{y}\|$.

(ii) $g_t(\mathbf{x}) \le G_t(\mathbf{x}) \le g_t(\mathbf{x}) + \frac{5}{4} p \rho \sqrt{d}$.

(iii) For some fixed $p \in \mathbb{Z}^+$, $\forall\, i \in [p]$, $\|\nabla^i G_t(\mathbf{x}) - \nabla^i G_t(\mathbf{y})\| \le \left(\frac{2}{\rho}\right)^i \|\mathbf{x} - \mathbf{y}\|$.

*3. Adding Uniform Convexity.* Now that the constructed function $G_t(\mathbf{x})$ is all-order smooth, we add to it the uniformly convex regularization. We define

$$f_t(\mathbf{x}) = \beta G_t(\mathbf{x}) \qquad\qquad\qquad f(\mathbf{x}) = f_T(\mathbf{x})$$

$$F_t(\mathbf{x}) = f_t(\mathbf{x}) + d_q(\mathbf{x}) \quad \text{for} \quad d_q(\mathbf{x}) = \frac{\sigma}{q}\|\mathbf{x}\|^q, \quad \mathbf{x} \in \mathcal{Q} \qquad F(\mathbf{x}) = F_T(\mathbf{x}),$$

where $\beta > 0$ is a parameter that we will choose later, $\mathcal{Q} = \{\mathbf{x} : \|\mathbf{x}\|_2 \le D\}$[1] for $D \le \left(\frac{H}{2^{1-\nu}C}\right)^{\frac{1}{q-p-\nu}}$ and $C = \sigma(q-1) \times \cdots \times (q-p)$.

**Lemma 4.** For $F(\mathbf{x}) = f_T(\mathbf{x}) + d_q(\mathbf{x})$ where $d_q(\mathbf{x}) = \frac{\sigma}{q}\|\mathbf{x}\|^q$ and $\mathbf{x} \in \mathcal{Q}$,

(i) $F$ is uniformly convex function with degree $q$ and modulus $\sigma > 0$.

(ii) $F(\mathbf{x})$ is $p^{th}$-order Hölder smooth with parameter $H = \frac{2^{p+1}\beta}{\rho^{p+\nu-1}}$, $\forall\, p \in \mathbb{Z}^+$.

Therefore, by Lemma 4, the function constructed satisfies the desired uniform convexity and high-order smoothness conditions. Next, we characterize with Lemma 5 the upper and lower bounds of the constructed function which will be used in the proof later.

**Lemma 5.** For $R(\mathbf{x}) = \beta \max_{k \in [T]} \xi_k \left\langle \mathbf{e}_{\alpha(k)}, \mathbf{x} \right\rangle + \frac{\sigma}{q}\|\mathbf{x}\|^q$, we have

$$R(\mathbf{x}) - \beta(T-1)\delta \le F(\mathbf{x}) \le R(\mathbf{x}) + \frac{5}{4} p \beta \rho \sqrt{d}.$$

### 4.2.2 Convergence Trajectory Generation

*4. Trajectory Generation Procedure.* The trajectory is generated following a standard $T$-step iterative procedure same as outlined in (Guzmán & Nemirovski, 2015; Doikov, 2022):

· For $t = 1$, $\mathbf{x}_1$ is the first point of the trajectory and is chosen by initialization of some algorithm $\mathcal{A}$, independent of $F$. Subsequently, choose

$$\alpha(1) \in \arg\max_{k \in [T]} \left|\left\langle \mathbf{e}_{\alpha(k)}, \mathbf{x}_1 \right\rangle\right| \qquad\qquad \xi_1 = \text{sign}\left(\left\langle \mathbf{e}_{\alpha(1)}, \mathbf{x}_1 \right\rangle\right),$$

after which a fixed $F_1(\mathbf{x})$ is generated.

---

[1] We would note that for the $q > p + \nu$ case, $F$ is guaranteed to be $p^{th}$-order smooth only in the bounded domain as constructed, since the regularization term $d_q(\mathbf{x})$ may not be $p^{th}$-order smooth on $\mathbb{R}^d$. The construction is inspired by that in (Juditsky & Nesterov, 2014). This is not explicitly discussed in (Song et al., 2021; Doikov, 2022; Thomsen & Doikov, 2024).

· For $2 \leq t \leq T$, at the beginning of each such iteration, we have access to $\mathbf{x}_1, \cdots, \mathbf{x}_{t-1}$, the function $F_{t-1}$, and its gradient information, which we denote as $\mathcal{I}_{t-1}(\mathbf{x}) = \{F_{t-1}, \nabla F_{t-1}, \cdots, \nabla^p F_{t-1}\}$. The algorithm $\mathcal{A}$ generates the next point with this information: $\mathbf{x}_t = \mathcal{A}(\mathcal{I}_{t-1}(\mathbf{x}_1), \cdots, \mathcal{I}_{t-1}(\mathbf{x}_{t-1}))$. Then choose

$$\alpha(t) \in \underset{k \in [T] \setminus \{\alpha(i): i < t\}}{\arg\max} \left| \langle \mathbf{e}_{\alpha(k)}, \mathbf{x}_t \rangle \right| \qquad \xi_t = \text{sign}\left( \langle \mathbf{e}_{\alpha(t)}, \mathbf{x}_t \rangle \right)$$

after which a fixed $F_t(\mathbf{x})$ is generated for the next iteration.

*5. Indistinguishability of $F_t$ and $F$ for Trajectory Generation.* It's important to note that the trajectory $\mathbf{x}_1, \cdots, \mathbf{x}_T$ is generated based on *a sequence of functions $F_1, \cdots, F_T$*, whereas our object of analysis should be just *one hard function $F = F_T$*. Here we show:

**Lemma 6.** *The trajectory $\mathbf{x}_1, \cdots, \mathbf{x}_T$ generated by applying an algorithm $\mathcal{A}$ iteratively on the sequence of functions $F_1, \cdots, F_T$, with up to $p^{th}$-order oracle access, is the same as the trajectory generated applying $\mathcal{A}$ directly on $F$ when oracle access pertains only local information within an $\ell_\infty$-ball with radius $\delta/2$.*

*Proof.* The idea is to show that $\forall\, 2 \leq t \leq T$, the function $g_t$ coincides with $g_T$ (so that $F_t$ coincides with $F_T$ in terms of generating $\mathbf{x}_{t+1}$, i.e., $\mathcal{I}_t = \mathcal{I}_T$) under some mild conditions. Similar proof can be found in (Guzmán & Nemirovski, 2015; Doikov, 2022, Section 3). By construction, $\forall\, t \in [T]$,

$$g_t(\mathbf{x}) = \max_{1 \leq k \leq t} r_k(\mathbf{x}) = \max\left\{ \max_{1 \leq k \leq s} r_k(\mathbf{x}), \max_{s < k \leq t} r_k(\mathbf{x}) \right\} = \max\left\{ g_s(\mathbf{x}), \max_{s < k \leq t} r_k(\mathbf{x}) \right\}$$

Furthermore, $\alpha(s) \in \arg\max_{k \in [T] \setminus \{\alpha(i): i < s\}} \left| \langle \mathbf{e}_{\alpha(k)}, \mathbf{x}_s \rangle \right|$ and $\xi_s = \text{sign}\left( \langle \mathbf{e}_{\alpha(s)}, \mathbf{x}_s \rangle \right)$, therefore

$$\begin{aligned}
g_s(\mathbf{x}_s) &= \max_{1 \leq k \leq s} \xi_k \langle \mathbf{e}_{\alpha(k)}, \mathbf{x}_s \rangle - (k-1)\delta \geq \max_{1 \leq k \leq s} \xi_k \langle \mathbf{e}_{\alpha(k)}, \mathbf{x}_s \rangle - (s-1)\delta \\
&\geq \left| \langle \mathbf{e}_{\alpha(s)}, \mathbf{x}_s \rangle \right| - (s-1)\delta \geq \max_{s < k \leq t} \xi_k \langle \mathbf{e}_{\alpha(k)}, \mathbf{x}_s \rangle - (s-1)\delta \\
&\geq \max_{s < k \leq t} \xi_k \langle \mathbf{e}_{\alpha(k)}, \mathbf{x}_s \rangle - (k-1)\delta + \delta \qquad (k, s \in \mathbb{Z}^+, k > s \implies k \geq s+1)
\end{aligned}$$

If we limit the information access within an $\ell_\infty$-ball with radius $\delta/2$ when searching for the next point $\mathbf{x}_{s+1}$ from $\mathbf{x}_s$, we then establish a local region $\forall \mathbf{x}, \|\mathbf{x} - \mathbf{x}_s\|_\infty \leq \frac{\delta}{2}$. Further by Lemma 2 that $g_s$ (also $\xi_k \langle \mathbf{e}_{\alpha(k)}, \mathbf{x} \rangle$) is 1-Lipschitz with respect to the $\ell_\infty$ norm, we have $\forall\, k$ such that $s < k \leq t$,

$$\begin{aligned}
g_s(\mathbf{x}_s) &\geq \xi_k \langle \mathbf{e}_{\alpha(k)}, \mathbf{x}_s \rangle - (k-1)\delta + 2\|\mathbf{x} - \mathbf{x}_s\|_\infty \\
&\geq \xi_k \langle \mathbf{e}_{\alpha(k)}, \mathbf{x}_s \rangle - (k-1)\delta + [g_s(\mathbf{x}_s) - g_s(\mathbf{x})] + \left[ \xi_k \langle \mathbf{e}_{\alpha(k)}, \mathbf{x} \rangle - \xi_k \langle \mathbf{e}_{\alpha(k)}, \mathbf{x}_s \rangle \right],
\end{aligned}$$

which implies that $g_s(\mathbf{x}) \geq \max_{s < k \leq t} \xi_k \langle \mathbf{e}_{\alpha(k)}, \mathbf{x} \rangle - (k-1)\delta = \max_{s < k \leq t} r_k(\mathbf{x})$. This concludes that $\forall\, \mathbf{x}$ such that $\|\mathbf{x} - \mathbf{x}_s\|_\infty \leq \frac{\delta}{2}$, $g_t(\mathbf{x}) = \max\{g_s(\mathbf{x}), \max_{s < k \leq t} r_k(\mathbf{x})\} = g_s(\mathbf{x})$, which further implies $F_t(\mathbf{x}) = F_s(\mathbf{x})$. Letting $t = T$ we have $\forall\, t \in [T]$, $F_t(\mathbf{x}) = F_T(\mathbf{x})$ for $\|\mathbf{x} - \mathbf{x}_t\|_\infty \leq \frac{\delta}{2}$. $\square$

### 4.2.3 LOWER BOUND DERIVATION

*6. Bounding the Optimality Gap.* The following lemma bounds optimality gap, whose proof is based on Lemma 5, and is presented in Appendix A.2.

**Lemma 7.** $F(\mathbf{x}_T) - F(\mathbf{x}^*) \geq -\beta(T-1)\delta - \frac{5}{4}p\beta\rho\sqrt{d} + \frac{q-1}{q}\left( \frac{\beta^q}{\sigma T^{\frac{q}{2}}} \right)^{\frac{1}{q-1}}.$

*7. Setting the parameters.* By Definition 4 and Lemma 14 (i), we know that $S_\rho[g_t](\mathbf{x}), \nabla S_\rho[g_t](\mathbf{x})$ depends on the value of $g_t(\mathbf{x})$ within an $\ell_\infty$-ball of radius $\rho$. Therefore inductively, we see that for $F(\mathbf{x}) = \beta S_\rho^p[g_T](\mathbf{x}) + \frac{\sigma}{q}\|\mathbf{x}\|^q$, $F(\mathbf{x}), \nabla F(\mathbf{x}), \cdots, \nabla^p F(\mathbf{x})$ depends on the value of $F(\mathbf{x})$ within an $\ell_\infty$-ball of radius $p\rho$. For our construction to hold, we also need $F_t(\mathbf{x})$ and $F(\mathbf{x})$ to be indistinguishable $\forall\, t \in [T]$, which is true within an $\ell_\infty$-ball of radius $\delta/2$.

Therefore, we set $\delta = 2p\rho$, so that for the purpose of oracle access at $\mathbf{x}_t$ (computing (high-order) gradients of $F$), it's indistinguishable to replace $F(\mathbf{x})$ with $F_t(\mathbf{x})$, and the sequence generated as in

Section 4.2.2 is the same as that directly applying some $p^{th}$-order algorithm $\mathcal{A}$ on $F(\mathbf{x})$. In other words, $F(\mathbf{x})$ and the generated $\mathbf{x}_T$ serve as valid components for deriving the lower bound.

As a result, $F(\mathbf{x}_T) - F(\mathbf{x}^*) \geq -2p\beta\rho(T-1+\frac{5}{8}\sqrt{d}) + \frac{q-1}{q}\left(\frac{\beta^q}{\sigma T^{\frac{q}{2}}}\right)^{\frac{1}{q-1}}$. Let $T \geq \sqrt{d}-1 \geq \frac{5}{8}\sqrt{d}-1$, then $F(\mathbf{x}_T) - F(\mathbf{x}^*) \geq -4p\beta\rho T + \frac{q-1}{q}\left(\frac{\beta^q}{\sigma T^{\frac{q}{2}}}\right)^{\frac{1}{q-1}}$. By letting $4p\beta\rho T = \frac{q-1}{2q}\left(\frac{\beta^q}{\sigma T^{\frac{q}{2}}}\right)^{\frac{1}{q-1}}$, we solve for $\rho = \frac{q-1}{8pq}\sigma^{-\frac{1}{q-1}}\beta^{\frac{1}{q-1}}T^{\frac{2-3q}{2(q-1)}} = c_q\sigma^{-\frac{1}{q-1}}\beta^{\frac{1}{q-1}}T^{\frac{2-3q}{2(q-1)}}$, in which $c_q = \frac{q-1}{8pq}$, and at the same time,

$$F(\mathbf{x}_T) - F(\mathbf{x}^*) \geq \frac{q-1}{2q}\left(\frac{\beta^q}{\sigma T^{\frac{q}{2}}}\right)^{\frac{1}{q-1}}. \tag{1}$$

By the construction of $F(\mathbf{x})$ and Lemma 4, we know that $F(\mathbf{x})$ is $p^{th}$-order Hölder smooth with parameter $H = \frac{2^{p+1}\beta}{\rho^{p+\nu-1}}$. Plugging in the value of $\rho$, we have

$$H = 2^{p+1}c_q^{-(p+\nu-1)}\sigma^{\frac{p+\nu-1}{q-1}}\beta^{-\frac{p-q+\nu}{q-1}}T^{\frac{(p+\nu-1)(3q-2)}{2(q-1)}},$$

equivalently, $\beta = \left(\frac{Hc_q^{p+\nu-1}\sigma^{-\frac{p+\nu-1}{q-1}}}{2^{p+1}}\right)^{-\frac{q-1}{p-q+\nu}}T^{\frac{(p+\nu-1)(3q-2)}{2(p-q+\nu)}}$. Plugging the value of $\beta$ back into Eq. (1), we have

$$F(\mathbf{x}_T) - F(\mathbf{x}^*) \geq \frac{q-1}{2q}\sigma^{-\frac{1}{q-1}}\left(\frac{Hc_q^{p+\nu-1}\sigma^{-\frac{p+\nu-1}{q-1}}}{2^{p+1}}\right)^{-\frac{q}{p-q+\nu}}T^{\frac{q[3(p+\nu)-2]}{2(p-q+\nu)}}$$

$$= 4p\left(\frac{q-1}{8pq}\right)^{\frac{(p+\nu)(1-q)}{p-q+\nu}}\sigma\left(\frac{H\sigma^{-1}}{2^{p+1}}\right)^{-\frac{q}{p-q+\nu}}T^{\frac{q[3(p+\nu)-2]}{2(p-q+\nu)}}.$$

We complete the proof for Theorem 1 by letting $F(\mathbf{x}_T) - F(\mathbf{x}^*) \leq \epsilon$, from which we solve for

$$T \geq \left(2^{(2p+2\nu+pq-q)/q}\right)^{-\frac{2}{3(p+\nu)-2}}p^{\frac{2(q-p-\nu)}{q[3(p+\nu)-2]}}\left(\frac{q-1}{8pq}\right)^{\frac{2(p+\nu)(q-1)}{q[3(p+\nu)-2]}}\left(\frac{H}{\sigma}\right)^{\frac{2}{3(p+\nu)-2}}\left(\frac{\sigma}{\epsilon}\right)^{\frac{2(q-p-\nu)}{q[3(p+\nu)-2]}}. \qquad \square$$

## 5 Lower Bound for the $q < p + \nu$ Case

**Theorem 2.** *For any $T$-step deterministic algorithm $\mathcal{A}$ with oracle access up to the $p^{th}$ order, there exists a convex function $f(\mathbf{x})$ whose $p^{th}$-order derivative is Hölder continuous of degree $\nu$ with modulus $H$ and a corresponding $F(\mathbf{x}) = f(\mathbf{x}) + \frac{\sigma}{q}\|\mathbf{x}\|^q$ with regularization that is uniformly convex of degree $q$ with modulus $\sigma$, such that $q < p + \nu$, it takes*

$$T \in \Omega\left(\left(\frac{H}{\sigma}\right)^{\frac{2}{3(p+\nu)-2}} + \log\log\left(\left(\frac{\sigma^{p+\nu}}{H^q}\right)^{\frac{1}{p+\nu-q}}\frac{1}{\epsilon}\right)\right)$$

*steps to reach an $\epsilon$-approximate solution $\mathbf{x}_T$ satisfying $F(\mathbf{x}_T) - F(\mathbf{x}^*) \leq \epsilon$.*

*Proof.* Similar to all other lower bound proofs, we construct such a function that satisfies the uniformly convex and Hölder smooth conditions and show that it requires at least the number of iterations stated in the theorem. The construction is generally based on Nesterov's hard function (Nesterov et al., 2018), and generalizes the construction in (Arjevani et al., 2019) to higher-order and the construction in (Kornowski & Shamir, 2020) to Hölder smooth functions as well as uniformly convex functions.

### 5.1 Function Construction based on Nesterov's Hard Function

A direct generalization of Nestrov's construction for first- and second-order lower bounds (Nesterov et al., 2018, Section 2.1.2, 4.3.1) to the $p^{th}$-order Hölder smooth setting takes the form $\tilde{f}(\mathbf{x}) = \frac{1}{p+\nu}\sum_{i=1}^{\tilde{T}}|x_i - x_{i+1}|^{p+\nu} - \gamma x_1 + \frac{\tilde{\sigma}}{q}\|\mathbf{x}\|^q$, for $q < p+\nu$, $\nu \in [0,1]$, which is uniformly convex by

the regularization. We further add a coefficient so that the function $p^{th}$-order Hölder smooth with the desired parameter $H$ and further on top of this a set of orthogonal basis $\mathbf{v}_i, \forall\, i \in [\tilde{T}]$ to limit the access of coordinates through the iterations:

$$f(\mathbf{x}) = \frac{H}{2^{p+\nu+1}(p+\nu-1)!} \left( \frac{1}{p+\nu} \sum_{i=1}^{\tilde{T}} |\langle \mathbf{v}_i, \mathbf{x} \rangle - \langle \mathbf{v}_{i+1}, \mathbf{x} \rangle|^{p+\nu} - \gamma \langle \mathbf{v}_1, \mathbf{x} \rangle \right) + \frac{\sigma}{q} \|\mathbf{x}\|^q,$$

for $\sigma = \frac{H\tilde{\sigma}}{2^{p+\nu+1}(p+\nu-1)!}$, or equivalently, $\tilde{\sigma} = \frac{2^{p+\nu+1}(p+\nu-1)!\sigma}{H}$. $\mathbf{v}_i$ is chosen iteratively to be orthogonal to $\mathbf{x}_1, \cdots, \mathbf{x}_i$ and $\mathbf{v}_1, \cdots, \mathbf{v}_{i-1}$. Similar to (Arjevani et al., 2019, Lemma 7), one can show that the oracle information of $f(\mathbf{x}_i), \forall\, i \leq t$ does not depend on $\mathbf{v}_{t+1}, \cdots, \mathbf{v}_{\tilde{T}}$, so that the iterative construction of $\mathbf{v}_i$ is valid, i.e., does not affect the $\mathbf{x}_i$ generated running an algorithm on $f$. Now we characterize the relation between $\tilde{f}$ and $f$.

**Lemma 8.** $\mathbf{x}^* = \arg\min_{\mathbf{x}} f(\mathbf{x})$, $\mathbf{y} = \arg\min_{\mathbf{x}} \tilde{f}(\mathbf{x})$. (i) $\forall i \in [\tilde{T}]$, $\langle \mathbf{v}_i, \mathbf{x}^* \rangle = y_i$. (ii) $\|\mathbf{x}^*\| = \|\mathbf{y}\|$.

Next, we characterize the convexity and smoothness of the constructed function. Specifically, we can show with the proof in Appendix B that $f$ satisfies the following lemma.

**Lemma 9.** $f(\mathbf{x})$ is (i) uniformly convex with degree $q$ and parameter $\sigma$. (ii) $p^{th}$-order Hölder smooth with degree $\nu$ and parameter $H$.

The analysis of (Nesterov et al., 2018) then derives the lower bound based on the closed-form optimal solution that minimizes the hard function. For our generalized construction of $f$, however, the closed-form solution is hard to obtain. As in (Arjevani et al., 2019), we instead analyze some properties of $f$ for each of these lower bounds. For simplicity, we state the properties for function $\tilde{f}$, and since $f$ is simply a scaling of $\tilde{f}$, the properties also apply to $f$ with a difference of constants. To prove Theorem 2, we show separately for the $\left( \frac{H}{\sigma} \right)^{\frac{2}{3(p+\nu)-2}}$ term and the $\log\log\left( \left( \frac{\sigma^{p+\nu}}{H^q} \right)^{\frac{1}{p+\nu-q}} \frac{1}{\epsilon} \right)$ term. The derivation is largely based on some key lemmas whose complete proof is in Appendix B.

## 5.2 $T \in \Omega\left( \left( \frac{H}{\sigma} \right)^{\frac{2}{3(p+\nu)-2}} \right)$

Since we cannot solve for a closed form solution from $\arg\min_{\mathbf{x}} \tilde{f}(\mathbf{x})$, we need to alternatively bound the solution in a relative scale. One key observation is that the coordinates of the optimal solution form a decreasing sequence (Arjevani et al., 2019; Carmon et al., 2021), and their relative relation can be characterized as in Lemma 10 (i) utilizing the first-order optimality condition. Based on the properties of each coordinate, one can relate them to the norm of the optimal solution as in Lemma 10 (iii).

**Lemma 10.** For $\mathbf{y} = \arg\min_{\mathbf{x}} \tilde{f}(\mathbf{x})$,

(i) $\forall t \in [\tilde{T}]$, $y_t \geq y_1 - (t-1)\gamma^{\frac{1}{p+\nu-1}}$.

(ii) For $\tilde{T} = \left\lfloor \frac{y_1}{\gamma^{\frac{1}{p+\nu-1}}} + 1 \right\rfloor$, $y_1 \leq \gamma^{\frac{1}{p+\nu-1}} + \sqrt{\frac{2\gamma^{\frac{p+\nu}{p+\nu-1}}}{\tilde{\sigma}\|\mathbf{y}\|^{q-2}}}$.

(iii) For $\gamma \geq \tilde{\sigma}^{\frac{p+\nu-1}{p+\nu-2}} \|\mathbf{y}\|^{\frac{(p+\nu-1)(q-2)}{p+\nu-2}}$, $\forall t \in [\tilde{T}]$, $y_t \geq \frac{\gamma^{\frac{p+\nu}{2(p+\nu-1)}}}{2^{p+\nu+1}\tilde{\sigma}^{\frac{1}{2}}\|\mathbf{y}\|^{\frac{q-2}{2}}} + \left( \frac{1}{2} - i \right)\gamma^{\frac{1}{p+\nu-1}}$.

Then the bound on the norm of the optimal solution can be established in the following lemma.

**Lemma 11.** $\|\mathbf{y}\| \leq \frac{2^{\frac{2}{3q-2}}\gamma^{\frac{3(p+\nu)-2}{(p+\nu-1)(3q-2)}}}{\tilde{\sigma}^{\frac{3}{3q-2}}}$.

The final step is to relate this norm to the optimality gap with the property of uniform convexity. By Lemma 8 and Lemma 10 (iii), $\langle \mathbf{v}_T, \mathbf{x}^* \rangle = y_T \geq \frac{\gamma^{\frac{p+\nu}{2(p+\nu-1)}}}{2^{p+\nu+1}\tilde{\sigma}^{\frac{1}{2}}\|\mathbf{y}\|^{\frac{q-2}{2}}} + \left( \frac{1}{2} - T \right)\gamma^{\frac{1}{p+\nu-1}}$. Therefore,

with $\mathbf{v}_T$ and $\mathbf{x}_T$ orthogonal to each other by construction,

$$f(\mathbf{x}_T) - f(\mathbf{x}^*) \geq \frac{\sigma}{q} \|\mathbf{x}_T - \mathbf{x}^*\|^q = \frac{\sigma}{q} \left( \sum_{i=1}^{\tilde{T}} (\langle \mathbf{v}_i, \mathbf{x}_T - \mathbf{x}^* \rangle)^2 \right)^{\frac{q}{2}} \geq \frac{\sigma}{q} \left( (\langle \mathbf{v}_T, \mathbf{x}_T - \mathbf{x}^* \rangle)^2 \right)^{\frac{q}{2}}$$

$$= \frac{\sigma}{q} (\langle \mathbf{v}_T, \mathbf{x}^* \rangle)^q \geq \frac{\sigma}{q} \left( \frac{\gamma^{\frac{p+\nu}{2(p+\nu-1)}}}{2^{p+\nu+1} \tilde{\sigma}^{\frac{1}{2}} \|\mathbf{y}\|^{\frac{q-2}{2}}} + \left( \frac{1}{2} - T \right) \gamma^{\frac{1}{p+\nu-1}} \right)^q$$

In order to achieve $f(\mathbf{x}_T) - f(\mathbf{x}^*) \leq \epsilon$, we have $\frac{\sigma}{q} \left( \frac{\gamma^{\frac{p+\nu}{2(p+\nu-1)}}}{2^{p+\nu+1} \tilde{\sigma}^{\frac{1}{2}} \|\mathbf{y}\|^{\frac{q-2}{2}}} + \left( \frac{1}{2} - T \right) \gamma^{\frac{1}{p+\nu-1}} \right)^q \leq \epsilon$,

from which we can solve for $T \geq \frac{\gamma^{\frac{p+\nu-2}{2(p+\nu-1)}}}{2^{p+\nu+1} \tilde{\sigma}^{\frac{1}{2}} \|\mathbf{y}\|^{\frac{q-2}{2}}} + \frac{1}{2} - \left( \frac{q\epsilon}{\sigma \gamma^{\frac{q}{p+\nu-1}}} \right)^{\frac{1}{q}}$. For $\epsilon \leq \frac{\gamma^{\frac{q}{p+\nu-1}} \sigma}{2^q q}$, we

have $\frac{1}{2} - \left( \frac{q\epsilon}{\sigma \gamma^{\frac{q}{p+\nu-1}}} \right)^{\frac{1}{q}} \geq 0$. Therefore, $T \geq \frac{\gamma^{\frac{p+\nu-2}{2(p+\nu-1)}}}{2^{p+\nu+1} \tilde{\sigma}^{\frac{1}{2}} \|\mathbf{y}\|^{\frac{q-2}{2}}}$.

By Lemma 11, we know that $\|\mathbf{y}\| \leq \frac{2^{\frac{2}{3q-2}} \gamma^{\frac{3(p+\nu)-2}{(p+\nu-1)(3q-2)}}}{\tilde{\sigma}^{\frac{3}{3q-2}}}$. Therefore, for $\mathbf{x}_0 = \mathbf{0}$, by Lemma 8,

$\|\mathbf{x}_0 - \mathbf{x}^*\| = \|\mathbf{x}^*\| = \|\mathbf{y}\| \leq \frac{2^{\frac{2}{3q-2}} \gamma^{\frac{3(p+\nu)-2}{(p+\nu-1)(3q-2)}}}{\tilde{\sigma}^{\frac{3}{3q-2}}}$. To satisfy the condition $\|\mathbf{x}_0 - \mathbf{x}^*\| \leq D$, we let

$\frac{2^{\frac{2}{3q-2}} \gamma^{\frac{3(p+\nu)-2}{(p+\nu-1)(3q-2)}}}{\tilde{\sigma}^{\frac{3}{3q-2}}} \leq D$, then we can solve for $\gamma \leq 2^{-\frac{2(p+\nu-1)}{3(p+\nu)-2}} D^{\frac{(p+\nu-1)(3q-2)}{3(p+\nu)-2}} \tilde{\sigma}^{\frac{3(p+\nu-1)}{3(p+\nu)-2}}$. Plug this

as well as $\|\mathbf{y}\| \leq D$ into the lower bound on $T$ we have

$$T \geq \frac{\left( 2^{-\frac{2(p+\nu-1)}{3(p+\nu)-2}} D^{\frac{(p+\nu-1)(3q-2)}{3(p+\nu)-2}} \tilde{\sigma}^{\frac{3(p+\nu-1)}{3(p+\nu)-2}} \right)^{\frac{p+\nu-2}{2(p+\nu-1)}}}{2^{p+\nu+1} \tilde{\sigma}^{\frac{1}{2}} D^{\frac{q-2}{2}}} = 2^{-\frac{p+\nu-2}{3(p+\nu)-2} - (p+\nu+1)} D^{\frac{2(p+\nu-q-1)}{3(p+\nu)-2}} \tilde{\sigma}^{-\frac{2}{3(p+\nu)-2}}$$

Plugging in $\tilde{\sigma} = \frac{2^{p+\nu+1}(p+\nu-1)! \sigma}{H}$, we have $T \in \Omega \left( \left( \frac{H}{\sigma} \right)^{\frac{2}{3(p+\nu)-2}} \right)$.

## 5.3 $T \in \Omega \left( \log \log \left( \left( \frac{\sigma^{p+\nu}}{H^q} \right)^{\frac{1}{p+\nu-q}} \frac{1}{\epsilon} \right) \right)$

For the $\log \log$ term, we follow a similar narrative as in Section 5.2, starting from characterizing the per-coordinate relation of the optimal solution.

**Lemma 12.** *For* $\mathbf{y} = \arg\min_{\mathbf{x}} \tilde{f}(\mathbf{x})$, *let* $t_1 \in [\tilde{T}]$ *be such that* $y_{t_1} > \frac{1}{p+\nu-1} \tilde{\sigma}^{\frac{1}{p+\nu-2}} \|\mathbf{y}\|^{\frac{q-2}{p+\nu-2}}$ *and* $y_{t_1+1} \leq \frac{1}{p+\nu-1} \tilde{\sigma}^{\frac{1}{p+\nu-2}} \|\mathbf{y}\|^{\frac{q-2}{p+\nu-2}}$. *Then*

(i) $\forall i \in [\tilde{T}], y_i = y_{i+1} + \left( \tilde{\sigma} \|\mathbf{y}\|^{q-2} \sum_{j=i+1}^{\tilde{T}} y_j \right)^{\frac{1}{p+\nu-1}}$ *and* $y_{i+1} \leq \frac{1}{\tilde{\sigma} \|\mathbf{y}\|^{q-2}} y_i^p$

(ii) $\forall i \geq t_1, \left( \frac{1}{c_{p,\nu}} \right)^{p+\nu-1} \frac{1}{\tilde{\sigma} \|\mathbf{y}\|^{q-2}} y_i^{p+\nu-1} \leq y_{i+1}$ *where* $c_{p,\nu}$ *is a constant depending on* $p, \nu$.

(iii) $\forall i \leq \tilde{T} - t_1, y_{t_1+i} \geq \left( \frac{1}{c_{p,\nu}} \right)^{\frac{(p+\nu-1)((p+\nu-1)^i-1)}{p+\nu-2}} \left( \tilde{\sigma} \|\mathbf{y}\|^{q-2} \right)^{\frac{1}{p+\nu-2}} (p+\nu-1)^{-(p+\nu-1)^i}$.

Next, we derive the bound on the norm $\|\mathbf{x}_T - \mathbf{x}^*\|^q$ from the coordinate-wise properties in Lemma 12 with the basis defined for $f$. When constructing the function, we choose $H \geq 2^{p+\nu+1}(p+\nu-1)!\sigma$ so that $\tilde{\sigma} \leq 1$. Then for basis vector $\mathbf{v}_{t_1+i}$, by Lemma 8 and Lemma 12 (iii),

$$\langle \mathbf{v}_{t_1+i}, \mathbf{x}^* \rangle = y_{t_1+i} \geq \left( \frac{1}{c_{p,\nu}} \right)^{\frac{(p+\nu-1)((p+\nu-1)^i-1)}{p+\nu-2}} \cdot \tilde{\sigma}^{\frac{1}{p+\nu-q}} \|\mathbf{y}\|^{\frac{q-2}{p+\nu-2}} \cdot (p+\nu-1)^{-(p+\nu-1)^i}$$

$$= \left( \frac{1}{c_{p,\nu}} \right)^{\frac{(p+\nu-1)((p+\nu-1)^i-1)}{p+\nu-2}} \cdot \left( \frac{2^{p+\nu+1}(p+\nu-1)!\sigma}{H} \right)^{\frac{1}{p+\nu-q}} \|\mathbf{x}_0 - \mathbf{x}^*\|^{\frac{q-2}{p+\nu-2}} \cdot (p+\nu-1)^{-(p+\nu-1)^i},$$

for $\mathbf{x}_0 = 0$, in which the first inequality follows from Lemma 12 (iii), and then the fact that for $q \geq 2$ and $\tilde{\sigma} \leq 1$, $\tilde{\sigma}^{\frac{1}{p+\nu-q}} \leq \tilde{\sigma}^{\frac{1}{p+\nu-2}}$. For $t_1 \leq \frac{\tilde{T}}{2}$,

$$\|\mathbf{x}_T - \mathbf{x}^*\|^q = (\|\mathbf{x}_T - \mathbf{x}^*\|^2)^{\frac{q}{2}} \geq \left( \sum_{i=1}^{\tilde{T}} (\langle \mathbf{v}_i, \mathbf{x}_T - \mathbf{x}^* \rangle)^2 \right)^{\frac{q}{2}} \geq \left( (\langle \mathbf{v}_{t_1+T}, \mathbf{x}_T - \mathbf{x}^* \rangle)^2 \right)^{\frac{q}{2}}$$

$$= \left( (\langle \mathbf{v}_{t_1+T}, \mathbf{x}^* \rangle)^2 \right)^{\frac{q}{2}} = (\langle \mathbf{v}_{t_1+T}, \mathbf{x}^* \rangle)^q$$

where the equality in the second line follows from the fact that by construction, $\mathbf{v}_{t_1+T}$ and $\mathbf{x}_T$ are orthogonal.

Finally, with uniform convexity, we have

$$f(\mathbf{x}_T) - f(\mathbf{x}^*) \geq \frac{\sigma}{q} \|\mathbf{x}_T - \mathbf{x}^*\|^q$$

$$\geq \left( \frac{1}{c_{p,\nu}} \right)^{\frac{q(p+\nu-1)((p+\nu-1)^T-1)}{p+\nu-2}} \cdot \frac{\sigma}{q} \left( \frac{2^{p+\nu+1}(p+\nu-1)!\sigma}{H} \right)^{\frac{q}{p+\nu-q}} \|\mathbf{x}_0 - \mathbf{x}^*\|^{\frac{q(q-2)}{p+\nu-2}} \cdot (p+\nu-1)^{-q(p+\nu-1)^T}$$

$$= c_{p,q,\nu} \cdot \frac{\sigma^{\frac{p+\nu}{p+\nu-q}}}{L_p^{\frac{q}{p+\nu-q}}} \cdot (p+\nu-1)^{-q(p+\nu-1)^T}$$

for $c_{p,q,\nu} = \frac{2^{\frac{(p+\nu+1)q}{p+\nu-q}}((p+\nu-1)!)^{\frac{q}{p+\nu-q}}}{q} \left( \frac{1}{c_{p,\nu}} \right)^{\frac{q(p+\nu-1)((p+\nu-1)^T-1)}{p+\nu-2}} D^{\frac{q(q-2)}{p+\nu-2}}$ in which $\|\mathbf{x}_0 - \mathbf{x}^*\| \leq D$.

In order to achieve $f(\mathbf{x}_T) - f(\mathbf{x}^*) \leq \epsilon$, we have $c_{p,q,\nu} \cdot \frac{\sigma^{\frac{p+\nu}{p+\nu-q}}}{L_p^{\frac{q}{p+\nu-q}}} \cdot (p+\nu-1)^{-q(p+\nu-1)^T} \leq \epsilon$, from

which we solve for $T \geq \log\log_{p+\nu-1} \left( c_{p,q,\nu} \frac{\sigma^{\frac{p+q-1}{p-1}}}{L_p^{\frac{q}{p-1}}} \cdot \frac{1}{\epsilon} \right) + \log_{p+\nu-1} \left( \frac{1}{q} \right)$, which completes the

proof for Theorem 2 combined with the result in Section 5.2. □

# 6 CONCLUSION AND FUTURE WORK

We provide tight lower bounds for minimizing functions with asymmetric high-order Hölder smoothness and uniform convexity. Specifically, we show that the oracle complexity is lower bounded by $\Omega\left( \left(\frac{H}{\sigma}\right)^{\frac{2}{3(p+\nu)-2}} \left(\frac{\sigma}{\epsilon}\right)^{\frac{2(q-p-\nu)}{q(3(p+\nu)-2)}} \right)$ for the $q > p + \nu$ case with the construction of a $\ell_\infty$-ball-truncated-Gaussian smoothed hard function, and $\Omega\left( \left(\frac{H}{\sigma}\right)^{\frac{2}{3(p+\nu)-2}} + \log\log\left( \left(\frac{\sigma^{p+\nu}}{H^q}\right)^{\frac{1}{p+\nu-q}} \frac{1}{\epsilon} \right) \right)$ for the $q < p + \nu$ case. Both lower bounds match the corresponding upper bounds in the general setting.

We note that the lower bounds for the $q > p + \nu$ case and the $q < p + \nu$ case are derived based on two different frameworks. The first lower bound based on Nemirovski's max function can be directly extended to hold for randomized algorithms based on "robust-zero-chain" arguments by (Carmon et al., 2020; 2021). The second lower bound based on Nesterov's function, which is not a robust zero-chain, holds only for deterministic/zero-respecting algorithms.

We further note that the lower bound for the $q = p + \nu$ case is not included in this paper. Proposing a unified framework for all three cases as well as generalizing the results to work for randomized algorithms would be of great interest, which we leave for future work.

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

# Appendices

## A  PROOF FOR TECHNICAL LEMMAS IN SECTION 4

### A.1  PROPERTIES OF TRUNCATED GAUSSIAN SMOOTHING

**Lemma 13** ($\ell_\infty$-Ball Truncated Gaussian and Its Marginal Distribution). *For standard MVN truncated in a unit $\ell_\infty$-ball as defined in Definition 4:*

$$\mathbb{P}[V = \mathbf{v}] = \frac{1}{Z(2\pi)^{\frac{d}{2}}} \exp\left\{-\frac{\mathbf{v}^\top \mathbf{v}}{2}\right\} \mathbb{I}_{[\|\mathbf{v}\|_\infty \leq 1]},$$

*(i) The cumulative distribution within the $\ell_\infty$-ball, i.e., the normalizing factor $Z(d) = [\Phi(1) - \Phi(-1)]^d$.*
*(ii) The marginal distribution is a standard normal truncated within $[-1, 1]$.*

*Proof.* (i) By Eq. (3) in (Cartinhour, 1990), we know that

$$\begin{aligned}
Z(d) &= \int_{\|\mathbf{v}\|_\infty \leq 1} \frac{1}{(2\pi)^{\frac{d}{2}}} \exp\left\{-\frac{\mathbf{v}^\top \mathbf{v}}{2}\right\} d\mathbf{v} \\
&= \underbrace{\int_{-1}^{1} \cdots \int_{-1}^{1}}_{d\text{-time integration, one for each coordinate}} \frac{1}{(2\pi)^{\frac{d}{2}}} \exp\left\{-\frac{\sum_{i=1}^{d} v_i^2}{2}\right\} dv_1 \cdots dv_d \\
&= \prod_{i=1}^{d} \int_{-1}^{1} \frac{1}{\sqrt{2\pi}} \exp\left\{-\frac{v_i^2}{2}\right\} dv_i \\
&= [\Phi(1) - \Phi(-1)]^d.
\end{aligned}$$

(ii) By Eq. (4) and (16) in (Cartinhour, 1990), $\forall i \in [d]$,

$$\begin{aligned}
\mathbb{P}[V_i] &= \frac{\exp\left\{-\frac{v_i^2}{2}\right\}}{[\Phi(1) - \Phi(-1)]^d \sqrt{2\pi}} \underbrace{\int_{-1}^{1} \cdots \int_{-1}^{1}}_{d-1\text{-time integration}} \frac{\exp\left\{-\frac{\sum_{j \neq i} v_j^2}{2}\right\}}{(2\pi)^{\frac{d-1}{2}}} dv_1 \cdots dv_{i-1} dv_{i+1} \cdots dv_d \\
&= \frac{\exp\left\{-\frac{v_i^2}{2}\right\}}{[\Phi(1) - \Phi(-1)]^d \sqrt{2\pi}} \prod_{j \neq i} \int_{-1}^{1} \frac{1}{\sqrt{2\pi}} \exp\left\{-\frac{v_j^2}{2}\right\} dv_j \\
&= \frac{\exp\left\{-\frac{v_i^2}{2}\right\}}{[\Phi(1) - \Phi(-1)]^d \sqrt{2\pi}} [\Phi(1) - \Phi(-1)]^{d-1} \\
&= \frac{1}{[\Phi(1) - \Phi(-1)] \sqrt{2\pi}} \exp\left\{-\frac{v_i^2}{2}\right\} \\
&= \frac{1}{\sqrt{2\pi} \int_{-1}^{1} \frac{1}{\sqrt{2\pi}} \exp\left\{-\frac{v_i^2}{2}\right\} dv_i} \exp\left\{-\frac{v_i^2}{2}\right\},
\end{aligned}$$

if $-1 \leq V_i \leq 1$, otherwise $\mathbb{P}[V_i] = 0$. Therefore, $V_i$ follows the truncated standard normal distribution within $[-1, 1]$. $\qquad\square$

**Lemma 14** (Properties of Truncated Gaussian Smoothing). *For a function $f : \mathbb{R}^d \to \mathbb{R}$ that is L-Lipschitz with respect to the $\ell_2$ norm, then $\forall \mathbf{x} \in \mathbb{R}^d$,*

*(i) If $f$ is convex and non-differentiable in a set with Lebesgue measure 0, then $f_\rho$ is continuously differentiable and $\nabla f_\rho(\mathbf{x}) = \mathbb{E}_V[\partial f(\mathbf{x} + \rho V)]$ for some random variable $V$.*
*(ii) If $f$ is convex, $f_\rho$ is convex and L-Lipschitz with respect to the $\ell_2$ norm.*

*(iii) If $f$ is convex, $f(\mathbf{x}) \leq f_\rho(\mathbf{x}) \leq f(\mathbf{x}) + \frac{5}{4}L\rho\sqrt{d}$.*
*(iv) $\nabla f_\rho$ is $\frac{2}{\rho}L$-Lipschitz, i.e., $f_\rho$ is $\frac{2}{\rho}L$-smooth.*

*Proof.* The proof of this lemma is based on that of Lemma 9 in (Duchi et al., 2012).

(i) The differentiability is established in (Bertsekas, 1973, Proposition 2.3), and $\nabla f_\rho(\mathbf{x}) = \mathbb{E}_V[\partial f(\mathbf{x} + \rho V)]$ in (Bertsekas, 1973, Proposition 2.2).

(ii) Expectation preserves convexity (Boyd & Vandenberghe, 2004, Section 3.2.1), therefore, given that $f$ is convex, by definition, $f_\rho$ is also convex. For Lipschitz continuity, by the second part of (i) and Jensen's inequality, we have

$$\|\nabla f_\rho(\mathbf{x})\| = \|\mathbb{E}_V[\partial f(\mathbf{x} + \rho V)]\|$$
$$\leq \mathbb{E}_V[\|\partial f(\mathbf{x} + \rho V)\|].$$

Given that $f$ is $L$-Lipschitz over $\mathbb{R}^d$ with respect to the $\ell_2$ norm, it is implied that $\forall \mathbf{x} \in \mathbb{R}^d$, $\|\partial f(\mathbf{x})\| \leq L$. As a result, $\|\nabla f_\rho(\mathbf{x})\| \leq \mathbb{E}[L] \leq L$ which further implies that $f_\rho$ is $L$-Lipschitz with respect to the $\ell_2$ norm.

(iii) For $f_\rho(\mathbf{x}) = \mathbb{E}_V[f(\mathbf{x} + \rho V)]$, $\mathbb{E}_V[V] = 0$ by construction. And since smoothing preserves convexity, $f_\rho(\mathbf{x})$ is also convex. For the lower bound, using Jensen's inequality,

$$f(\mathbf{x}) = f(\mathbf{x} + \rho \mathbb{E}_V[V])$$
$$= f(\mathbb{E}_V[\mathbf{x} + \rho V])$$
$$\leq \mathbb{E}_V[f(\mathbf{x} + \rho V)]$$
$$= f_\rho(\mathbf{x}).$$

For the upper bound, since $f$ is $L$-Lipschitz in $\ell_2$-norm, $f(\mathbf{x} + \rho V) - f(\mathbf{x}) \leq L\|\rho V\|$. Therefore,

$$f_\rho(\mathbf{x}) = \mathbb{E}_V[f(\mathbf{x} + \rho V)]$$
$$\leq \mathbb{E}_V[f(\mathbf{x}) + L\rho\|V\|]$$
$$= f(\mathbf{x}) + L\rho \mathbb{E}\left[\sqrt{\sum_{i=1}^d V_i^2}\right]$$
$$\leq f(\mathbf{x}) + L\rho \sqrt{\sum_{i=1}^d \mathbb{E}[V_i^2]}.$$

By Lemma 13 (ii), $V_i$ follows the standard normal distribution truncated within $[-1, 1]$. Therefore, let $\Phi(\cdot)$ denote the cumulative distribution function of standard normal distribution, then

$$\mathbb{E}[V_i^2] = \int_{-1}^1 \frac{\phi(\tau)}{\Phi(1) - \Phi(-1)} \tau^2 d\tau$$
$$= \frac{1}{\Phi(1) - \Phi(-1)} \int_{-1}^1 \phi(\tau)\tau^2 d\tau$$
$$\leq \frac{1}{\Phi(1) - \Phi(-1)} \int_{-\infty}^\infty \phi(\tau)\tau^2 d\tau$$
$$= \frac{\mathbb{E}[U_i^2]}{\Phi(1) - \Phi(-1)}$$

for $U_i \sim \mathcal{N}(0, 1)$, $\forall i \in [d]$. Then for $U = [U_1, \cdots, U_d]^\top$, $U$ follows the standard MVN distribution and

$$f_\rho(\mathbf{x}) \leq f(\mathbf{x}) + L\rho \sqrt{\frac{\mathbb{E}\left[\sum_{i=1}^d U_i^2\right]}{\Phi(1) - \Phi(-1)}}$$
$$= f(\mathbf{x}) + L\rho \sqrt{\frac{\mathbb{E}[\|U\|^2]}{\Phi(1) - \Phi(-1)}}.$$

$\mathbb{E}\left[\|U\|^2\right]$ is the second moment of the standard MVN, which is bounded by the dimension $d$ (Nesterov & Spokoiny, 2017, Lemma 1). We know that $\Phi(1) - \Phi(-1) \approx 0.6827$. As a result, we have

$$f_\rho(\mathbf{x}) \leq f(\mathbf{x}) + \frac{5}{4} L \rho \sqrt{d}.$$

(iv) The proof of this lemma follows that of Lemma 3.3 point 3 in (Lakshmanan & De Farias, 2008), also seen in that of Lemma 9 (iii) in (Duchi et al., 2012). Denote the PDF of the unit $\ell_\infty$-ball-truncated standard MVN as $\phi_{\|\cdot\|_\infty \leq 1}(\cdot; 0, 1)$. Then for $f_\rho(\mathbf{x}) = \mathbb{E}_V[f(\mathbf{x} + \rho V)]$, $\rho V$ has PDF $\phi_{\|\cdot\|_\infty \leq \rho}(\cdot; 0, \rho^2)$ by Lemma 2 (v) in (Chen et al., 2020). By (Duchi et al., 2012, Lemma 11), $\forall \mathbf{x}, \mathbf{x}' \in \mathbb{R}^d$, for $Z$ from $\phi_{\|\cdot\|_\infty \leq \rho}(\cdot; 0, \rho^2)$,

$$\|\nabla f_\rho(\mathbf{x}) - \nabla f_\rho(\mathbf{x}')\|_2 \leq L \underbrace{\int \left|\phi_{\|\cdot\|_\infty \leq \rho}(\mathbf{z} - \mathbf{x}; 0, \rho^2) - \phi_{\|\cdot\|_\infty \leq \rho}(\mathbf{z} - \mathbf{x}'; 0, \rho^2)\right| d\mathbf{z}}_{I}.$$

Now we bound the integral. Note that $\forall \mathbf{x}$, $\phi_{\|\cdot\|_\infty \leq \rho}(\mathbf{x}; 0, \rho^2)$ is a truncated MVN symmetrically centered at the origin, consequently, is strictly decreasing with respect to $\|\mathbf{x}\|_2^2$. As a result, $\phi_{\|\cdot\|_\infty \leq \rho}(\mathbf{z} - \mathbf{x}; 0, \rho^2) \geq \phi_{\|\cdot\|_\infty \leq \rho}(\mathbf{z} - \mathbf{x}'; 0, \rho^2)$ if and only if $\|\mathbf{z} - \mathbf{x}\|_2 \leq \|\mathbf{z} - \mathbf{x}'\|_2$. Therefore,

$$I = 2 \int_{\|\mathbf{z}-\mathbf{x}\|_2 \leq \|\mathbf{z}-\mathbf{x}'\|_2} \left(\phi_{\|\cdot\|_\infty \leq \rho}(\mathbf{z} - \mathbf{x}; 0, \rho^2) - \phi_{\|\cdot\|_\infty \leq \rho}(\mathbf{z} - \mathbf{x}'; 0, \rho^2)\right) d\mathbf{z}$$

$$= 2 \int_{\|\mathbf{z}-\mathbf{x}\|_2 \leq \|\mathbf{z}-\mathbf{x}'\|_2} \phi_{\|\cdot\|_\infty \leq \rho}(\mathbf{z} - \mathbf{x}; 0, \rho^2) d\mathbf{z} - 2 \int_{\|\mathbf{z}-\mathbf{x}\|_2 \leq \|\mathbf{z}-\mathbf{x}'\|_2} \phi_{\|\cdot\|_\infty \leq \rho}(\mathbf{z} - \mathbf{x}'; 0, \rho^2) d\mathbf{z}.$$

Denote $\mathbf{y} = \mathbf{z} - \mathbf{x}$ and $\mathbf{y}' = \mathbf{z} - \mathbf{x}'$, then

$$I = 2 \int_{\|\mathbf{y}\|_2 \leq \|\mathbf{y}-(\mathbf{x}'-\mathbf{x})\|_2} \phi_{\|\cdot\|_\infty \leq \rho}(\mathbf{y}; 0, \rho^2) d\mathbf{y} - 2 \int_{\|\mathbf{y}'\|_2 \geq \|\mathbf{y}'-(\mathbf{x}-\mathbf{x}')\|_2} \phi_{\|\cdot\|_\infty \leq \rho}(\mathbf{y}'; 0, \rho^2) d\mathbf{y}'$$

$$= 2\mathbb{P}_{\phi_{\|\cdot\|_\infty \leq \rho}}\left[\|Z\|_2 \leq \|Z - (\mathbf{x}' - \mathbf{x})\|_2\right] - 2\mathbb{P}_{\phi_{\|\cdot\|_\infty \leq \rho}}\left[\|Z'\|_2 \geq \|Z' - (\mathbf{x} - \mathbf{x}')\|_2\right]$$

$$= 2\mathbb{P}_{\phi_{\|\cdot\|_\infty \leq \rho}}\left[\|Z\|_2^2 \leq \|Z - (\mathbf{x}' - \mathbf{x})\|_2^2\right] - 2\mathbb{P}_{\phi_{\|\cdot\|_\infty \leq \rho}}\left[\|Z'\|_2^2 \geq \|Z' - (\mathbf{x} - \mathbf{x}')\|_2^2\right]$$

$$= 2\mathbb{P}_{\phi_{\|\cdot\|_\infty \leq \rho}}\left[2\langle Z, \mathbf{x}' - \mathbf{x}\rangle \leq \|\mathbf{x}' - \mathbf{x}\|_2^2\right] - 2\mathbb{P}_{\phi_{\|\cdot\|_\infty \leq \rho}}\left[2\langle Z', \mathbf{x} - \mathbf{x}'\rangle \geq \|\mathbf{x} - \mathbf{x}'\|_2^2\right]$$

$$= 2\mathbb{P}_{\phi_{\|\cdot\|_\infty \leq \rho}}\left[\left\langle Z, \frac{\mathbf{x}' - \mathbf{x}}{\|\mathbf{x}' - \mathbf{x}\|_2}\right\rangle \leq \frac{\|\mathbf{x}' - \mathbf{x}\|_2}{2}\right] - 2\mathbb{P}_{\phi_{\|\cdot\|_\infty \leq \rho}}\left[\left\langle Z', \frac{\mathbf{x} - \mathbf{x}'}{\|\mathbf{x} - \mathbf{x}'\|_2}\right\rangle \geq \frac{\|\mathbf{x} - \mathbf{x}'\|_2}{2}\right]$$

Denote $W = \left\langle Z, \frac{\mathbf{x}' - \mathbf{x}}{\|\mathbf{x}' - \mathbf{x}\|_2}\right\rangle$ and $W' = \left\langle Z', \frac{\mathbf{x} - \mathbf{x}'}{\|\mathbf{x} - \mathbf{x}'\|_2}\right\rangle$. Since $\frac{\mathbf{x}' - \mathbf{x}}{\|\mathbf{x}' - \mathbf{x}\|_2}$ and $\frac{\mathbf{x} - \mathbf{x}'}{\|\mathbf{x} - \mathbf{x}'\|_2}$ are normalized vectors, $W$ and $W'$ follow the one-dimensional distribution projected onto a plane along some direction from the truncated multivariate Gaussian, which is symmetrically centered at the origin. Therefore, by symmetry,

$$I = 2\mathbb{P}\left[W \leq \frac{\|\mathbf{x}' - \mathbf{x}\|_2}{2}\right] - 2\mathbb{P}\left[W' \geq \frac{\|\mathbf{x} - \mathbf{x}'\|_2}{2}\right]$$

$$= 2\mathbb{P}\left[W \leq -\frac{\|\mathbf{x}' - \mathbf{x}\|_2}{2}\right] + 2\mathbb{P}\left[-\frac{\|\mathbf{x}' - \mathbf{x}\|_2}{2} \leq W \leq \frac{\|\mathbf{x}' - \mathbf{x}\|_2}{2}\right] - 2\mathbb{P}\left[W' \geq \frac{\|\mathbf{x} - \mathbf{x}'\|_2}{2}\right]$$

$$= 2\mathbb{P}\left[-\frac{\|\mathbf{x}' - \mathbf{x}\|_2}{2} \leq W \leq \frac{\|\mathbf{x}' - \mathbf{x}\|_2}{2}\right]$$

As we later upper bound the integration by the peak of this distribution, we know by the geometry of $\ell_\infty$-ball that the projection onto the diagonal yields the highest peak, and that is when $W = \frac{1}{\sqrt{d}} \sum_{i=1}^d Z_i$ for $Z_i$ being the marginal of $Z$ that follows the truncated Gaussian distribution on $[-\rho, \rho]$ by Lemma 13 (ii). And further by Lemma 2 (v) in (Chen et al., 2020), $\frac{Z_i}{\sqrt{d}}$ is also a truncated Gaussian whose PDF is $\phi_{[-\frac{\rho}{\sqrt{d}}, \frac{\rho}{\sqrt{d}}]}(w; 0, \frac{\rho^2}{d})$. As a result, $W$ is the sum of independent identically

---

[2]Importantly, the PDF is strictly decreasing with respect to $\|\cdot\|_2$, not $\|\cdot\|_\infty$, no matter in which norm the truncation is done, as long as centered at the origin.

distributed (i.i.d.) truncated Gaussian variables. By Theorem 3 in (Chen et al., 2020) and E.q. (4.2) in (Birnbaum & Andrews, 1949) we know the sum of truncated Gaussian variables converges to a normal distribution for large $d$. As a result, $W \sim \left( \sum_{i=1}^{d} \text{Var}\,[Z_i] \right) \mathcal{N}\,(0,1)$. Knowing from the CDF of truncated Gaussian that $\forall\, i \in [d]$, $\text{Var}\,[Z_i] = \frac{\sigma^2}{d} \left[ 1 - \frac{\phi(1)+\phi(-1)}{\Phi(1)-\Phi(-1)} - \left( \frac{\phi(1)-\phi(-1)}{\Phi(1)-\Phi(-1)} \right)^2 \right] = 0.7089 \frac{\rho^2}{d}$, we have $\left( \sum_{i=1}^{d} \text{Var}\,[Z_i] \right) = 0.7089 \rho^2$

$$
\mathbb{P}\left[ -\frac{\|\mathbf{x}' - \mathbf{x}\|_2}{2} \le W \le \frac{\|\mathbf{x}' - \mathbf{x}\|_2}{2} \right] = \frac{1}{\sqrt{2\pi}\sqrt{0.7089}\rho} \int_{-\frac{\|\mathbf{x}'-\mathbf{x}\|_2}{2}}^{\frac{\|\mathbf{x}'-\mathbf{x}\|_2}{2}} \exp\{-\frac{w^2}{2 \times 0.7089\rho^2}\} dw
$$

Furthermore, since the PDF takes its peak at $w = 0$, we have

$$
I \le 2 \times \frac{2}{\sqrt{2\pi}\rho} \int_{-\frac{\|\mathbf{x}'-\mathbf{x}\|_2}{2}}^{\frac{\|\mathbf{x}'-\mathbf{x}\|_2}{2}} dw
$$

$$
= \frac{4\|\mathbf{x}' - \mathbf{x}\|_2}{\sqrt{2\pi}\rho}
$$

Therefore,

$$
\|\nabla f_\rho(\mathbf{x}) - \nabla f_\rho(\mathbf{x}')\|_2 \le LI
$$

$$
\le \frac{2L}{\rho}\|\mathbf{x}' - \mathbf{x}\|_2.
$$

$\square$

**Lemma 1.** *Given a L-Lipschitz function $f$, the function $f_\rho^p = S_\rho[\cdots [S_\rho[f]] \cdots ]$ satisfies*

(i) *If $f$ is convex, $f_\rho^p$ is convex and L-Lipschitz with respect to the $\ell_2$ norm.*

(ii) *If $f$ is convex, $f(\mathbf{x}) \le f_\rho^p(\mathbf{x}) \le f(\mathbf{x}) + \frac{5p}{4} L\rho\sqrt{d}$.*

(iii) *$\forall i \in [p]$, $\forall \mathbf{x}, \mathbf{x}' \in \mathbb{R}^d$, $\|\nabla^i f_\rho^p(\mathbf{x}) - \nabla^i f_\rho^p(\mathbf{x}')\| \le \left( \frac{2}{\rho} \right)^i L\|\mathbf{x} - \mathbf{x}'\|.$*

*Proof.* The proof of this lemma relies on inductively applying Lemma 14 and we provide formal proof by induction.

(i) The base case $p = 1$ holds directly by Lemma 14 (ii). Then we state the hypothesis that for $p = k$, $f_\rho^k$ is convex and $L$-Lipschitz with respect to the $\ell_2$ norm. For the induction step, we have, by definition, $f_\rho^{k+1} = S_\rho[f_\rho^k]$ where $f_\rho^k$ is convex and $L$-Lipschitz with respect to the $\ell_2$ norm by our hypothesis, with which $f_\rho^k$ satisfies the condition of Lemma 14. Then by Lemma 14 (ii), $f_\rho^{k+1}$ is convex and $L$-Lipschitz with respect to the $\ell_2$ norm.

(ii) The base case $p = 1$ holds directly by Lemma 14 (iii). Then we state the hypothesis that for $p = k$, $f(\mathbf{x}) \le f_\rho^k(\mathbf{x}) \le f(\mathbf{x}) + \frac{5k}{4} L\rho\sqrt{d}$ holds. From the result of (i), we know that $f_\rho^k$ satisfies the condition of Lemma 14. Therefore, applying 14 (iii) to the function $f_\rho^k(\mathbf{x})$ we have for the lower bound

$$
f_\rho^{k+1}(\mathbf{x}) \ge f_\rho^k(\mathbf{x}) \ge f(\mathbf{x})
$$

and for the lower bound

$$
f_\rho^{k+1}(\mathbf{x}) \le f_\rho^k(\mathbf{x}) + \frac{5}{4} L\rho\sqrt{d} \le f(\mathbf{x}) + \frac{5k}{4} L\rho\sqrt{d} + \frac{5}{4} L\rho\sqrt{d} = f(\mathbf{x}) + \frac{5(k+1)}{4} L\rho\sqrt{d}
$$

which completes the induction step.

(iii) The base case $p = 1$ holds for $i = 0$ by Lemma 14 (ii) and for $i = 1$ by Lemma 14 (iv). Now we state the inductive hypothesis that for $p = k$, it holds that $\forall\, \mathbf{x}, \mathbf{x}' \in \mathbb{R}$,

$$
\forall\, i \in [k], \ \|\nabla^i f_\rho^k(\mathbf{x}) - \nabla^i f_\rho^k(\mathbf{x}')\| \le \left( \frac{2}{\rho} \right)^i L\|\mathbf{x} - \mathbf{x}'\|.
$$

That is, $\forall\, i \in [k]$, the function $\nabla^i f_\rho^k$ is $\left( \left( \frac{2}{\rho} \right)^i L \right)$-Lipschitz. Then for $p = k+1$, $\forall i \in [k+1]$,

$$
\begin{aligned}
\| \nabla^i f_\rho^{k+1}(\mathbf{x}) - \nabla^i f_\rho^{k+1}(\mathbf{x}') \| &= \| \nabla^i S_\rho[f_\rho^k](\mathbf{x}) - \nabla^i S_\rho[f_\rho^k](\mathbf{x}') \| \\
&= \| S_\rho[\nabla^i f_\rho^k](\mathbf{x}) - S_\rho[\nabla^i f_\rho^k](\mathbf{x}') \| \\
&= \| \mathbb{E}_V[\nabla^i f_\rho^k(\mathbf{x} + \rho V)] - \mathbb{E}_V[\nabla^i f_\rho^k(\mathbf{x}' + \rho V)] \| \\
&= \| \mathbb{E}_V[\nabla^i f_\rho^k(\mathbf{x} + \rho V) - \nabla^i f_\rho^k(\mathbf{x}' + \rho V)] \| \\
&\leq \mathbb{E}_V[\| \nabla^i f_\rho^k(\mathbf{x} + \rho V) - \nabla^i f_\rho^k(\mathbf{x}' + \rho V) \|]
\end{aligned}
$$

where the first equality holds by definition, the second equality by the fact that expectation and derivative commute for differentiable functions, and the last inequality by the Jensen's.

For $i < k+1$, we can directly apply Lemma 14 (iv), with the hypothesis as the condition, on the function $\nabla^i f_\rho^k$, to establish the result that $\nabla^i f_\rho^k$ is smooth with parameter $\left( \frac{2}{\rho} \right)^i L$. Therefore,

$$
\begin{aligned}
\| \nabla^i f_\rho^{k+1}(\mathbf{x}) - \nabla^i f_\rho^{k+1}(\mathbf{x}') \| &\leq \mathbb{E}_V[\| \nabla^i f_\rho^k(\mathbf{x} + \rho V) - \nabla^i f_\rho^k(\mathbf{x}' + \rho V) \|] \\
&\leq \mathbb{E}_V\left[ \left( \frac{2}{\rho} \right)^i L \| \mathbf{x} - \mathbf{x}' \| \right] \\
&= \left( \frac{2}{\rho} \right)^i L \| \mathbf{x} - \mathbf{x}' \|
\end{aligned}
$$

For $i = k+1$, we have from our $i < k+1$ case that the function $\nabla^k f_\rho^{k+1}$ is $\left( \left( \frac{2}{\rho} \right)^k L \right)$-Lipschitz. We can therefore apply Lemma 14 (iv) on $\nabla^k f_\rho^{k+1}$ and claim that it's also smooth with parameter $\frac{2}{\rho} \cdot \left( \frac{2}{\rho} \right)^k L = \left( \frac{2}{\rho} \right)^{k+1} L$. That is,

$$
\left\| \nabla \left[ \nabla^k f_\rho^{k+1} \right] (\mathbf{x}) - \nabla \left[ \nabla^k f_\rho^{k+1} \right] (\mathbf{x}') \right\| \leq \left( \frac{2}{\rho} \right)^{k+1} L \| \mathbf{x} - \mathbf{x}' \|,
$$

which completes the proof. $\qquad\square$

## A.2 PROPERTIES OF THE CONSTRUCTED HARD FUNCTION

**Lemma 2.** $\forall\, t \in [T]$, $g_t$ is convex and 1-Lipschitz with respect to the $\ell_\infty$-norm, and also the $\ell_2$-norm.

*Proof.* (1) For convexity, by definition we have

$$
g_t(\mathbf{x}) = \max_{1 \leq k \leq t} r_k(\mathbf{x}) \qquad where \qquad \forall\, k \in [T], r_k(\mathbf{x}) = \xi_k \langle \mathbf{e}_{\alpha(k)}, \mathbf{x} \rangle - (k-1)\delta,
$$

Since $r_k(\mathbf{x})$ is linear in $\mathbf{x}$, $r_k(\mathbf{x})$ is convex. Then $g_t(\mathbf{x})$ is the maximum of convex functions which is also convex.

(2) To show Lipschitzness, $\forall\, \mathbf{x}, \mathbf{y} \in \mathbb{R}^d$, without the loss of generality, denote

$$
k_1 = \arg\max_{1 \leq k \leq t} r_k(\mathbf{x}) \qquad\qquad k_2 = \arg\max_{1 \leq k \leq t} r_k(\mathbf{y}).
$$

Therefore,

$$
g_t(\mathbf{x}) = \xi_{k_1} x_{\alpha(k_1)} - (k_1 - 1)\delta \qquad\qquad g_t(\mathbf{y}) = \xi_{k_2} y_{\alpha(k_2)} - (k_2 - 1)\delta.
$$

Since

$$
\begin{aligned}
g_t(\mathbf{y}) &= \xi_{k_2} y_{\alpha(k_2)} - (k_2 - 1)\delta \\
&= \max_{1 \leq k \leq t} \xi_k \langle \mathbf{e}_{\alpha(k)}, \mathbf{x} \rangle - (k-1)\delta \\
&\geq \xi_{k_1} y_{\alpha(k_1)} - (k_1 - 1)\delta,
\end{aligned}
$$

we have

$$
\begin{aligned}
g_t(\mathbf{x}) - g_t(\mathbf{y}) &\leq \left(\xi_{k_1} x_{\alpha(k_1)} - (k_1 - 1)\delta\right) - \left(\xi_{k_1} y_{\alpha(k_1)} - (k_1 - 1)\delta\right) \\
&\leq |x_{\alpha(k_1)} - y_{\alpha(k_1)}| \\
&\leq \max_{1 \leq i \leq d} |x_i - y_i| \\
&= \|\mathbf{x} - \mathbf{y}\|_\infty \\
&\leq \|\mathbf{x} - \mathbf{y}\|_2,
\end{aligned}
$$

where the last two inequalities show Lipschitzness in $\ell_\infty$ and $\ell_2$ norm respectively. $\quad\square$

**Lemma 3.** $\forall\, t \in [T], \forall\, \mathbf{x}, \mathbf{y} \in \mathbb{R}^d,$

(i) $G_t(\mathbf{x})$ is convex and 1-Lipschitz, i.e., $G_t(\mathbf{x}) - G_t(\mathbf{y}) \leq \|\mathbf{x} - \mathbf{y}\|.$

(ii) $g_t(\mathbf{x}) \leq G_t(\mathbf{x}) \leq g_t(\mathbf{x}) + \frac{5}{4} p \rho \sqrt{d}.$

(iii) For some fixed $p \in \mathbb{Z}^+, \forall\, i \in [p], \|\nabla^i G_t(\mathbf{x}) - \nabla^i G_t(\mathbf{y})\| \leq \left(\frac{2}{\rho}\right)^i \|\mathbf{x} - \mathbf{y}\|.$

*Proof.* The proof follows directly from that for Lemma 1. $\quad\square$

**Lemma 4.** For $F(\mathbf{x}) = f_T(\mathbf{x}) + d_q(\mathbf{x})$ where $d_q(\mathbf{x}) = \frac{\sigma}{q} \|\mathbf{x}\|^q$ and $\mathbf{x} \in \mathcal{Q},$

(i) $F$ is uniformly convex function with degree $q$ and modulus $\sigma > 0.$

(ii) $F(\mathbf{x})$ is $p^{th}$-order Hölder smooth with parameter $H = \frac{2^{p+1}\beta}{\rho^{p+\nu-1}}, \forall\, p \in \mathbb{Z}^+.$

*Proof.* (i) It is shown in (Nesterov et al., 2018, Section 4.2.2) that $\frac{\sigma}{q} \|\mathbf{x}\|^q$ is uniformly convex with degree $q$ and parameter $\sigma$. By Lemma 3 (i), $G_T$ is convex, therefore $f$ is also convex, so that $\forall\, \mathbf{x}, \mathbf{y} \in \mathbb{R}, \langle \nabla f(\mathbf{x}) - \nabla f(\mathbf{y}), \mathbf{x} - \mathbf{y} \rangle \geq 0$. Therefore, by Definition 3, $\left\langle \nabla(\frac{\sigma}{q} \|\mathbf{x}\|^q) - \nabla(\frac{\sigma}{q} \|\mathbf{y}\|^q), \mathbf{x} - \mathbf{y} \right\rangle \geq \sigma \|\mathbf{x} - \mathbf{y}\|^q$. Adding them together we get $\langle \nabla F(\mathbf{x}) - \nabla F(\mathbf{y}), \mathbf{x} - \mathbf{y} \rangle \geq \sigma \|\mathbf{x} - \mathbf{y}\|^q$, which shows that $F(\mathbf{x})$ is uniformly convex function with degree $q$ and modulus $\sigma > 0.$

(ii) From Lemma 3 (iii) and Definition 1, we know that $f$ is $p^{th}$-order smooth with parameter $L_p = \beta \left(\frac{2}{\rho}\right)^p, \forall\, p \in \mathbb{Z}^+$, i.e., $\forall\, \mathbf{x}, \mathbf{y} \in \mathcal{Q} \subset \mathbb{R}^d,$

$$
\|\nabla^p f(\mathbf{x}) - \nabla^p f(\mathbf{y})\| \leq \beta \left(\frac{2}{\rho}\right)^p \|\mathbf{x} - \mathbf{y}\|.
$$

Also, $\nabla^{p-1} f$ is $\beta \left(\frac{2}{\rho}\right)^{p-1}$-Lipschitz, which implies that $\forall\, \mathbf{x} \in \mathbb{R}^d, \|\nabla^p f(\mathbf{x})\| \leq \beta \left(\frac{2}{\rho}\right)^{p-1}$. Then we have $\forall\, \mathbf{x}, \mathbf{y} \in \mathcal{Q},$

$$
\begin{aligned}
\|\nabla^p f(\mathbf{x}) - \nabla^p f(\mathbf{y})\| &= \|\nabla^p f(\mathbf{x}) - \nabla^p f(\mathbf{y})\|^\nu \|\nabla^p f(\mathbf{x}) - \nabla^p f(\mathbf{y})\|^{1-\nu} \\
&\leq \|\nabla^p f(\mathbf{x}) - \nabla^p f(\mathbf{y})\|^\nu \left(\|\nabla^p f(\mathbf{x})\| + \|\nabla^p f(\mathbf{y})\|\right)^{1-\nu} \\
&\leq \left(\frac{2}{\rho}\right)^{p\nu} \beta^\nu \|\mathbf{x} - \mathbf{y}\|^\nu \left(2\beta \left(\frac{2}{\rho}\right)^{p-1}\right)^{1-\nu} \\
&= \frac{2^p \beta}{\rho^{p+\nu-1}} \|\mathbf{x} - \mathbf{y}\|^\nu.
\end{aligned}
$$

By letting $H = \frac{2^{p+1}\beta}{\rho^{p+\nu-1}}$, we can conclude that $f$ is $p^{th}$-order Hölder smooth with parameter $\frac{H}{2}.$

Furthermore, for $d_q(\mathbf{x})$, by definition, $\mathcal{Q} = \{\mathbf{x} : \|\mathbf{x}\|_2 \leq D\}$ for $D \leq \left(\frac{H}{2^{1-\nu}C}\right)^{\frac{1}{q-p-\nu}}$ and $C = \sigma(q-1) \times \cdots \times (q-p)$. As a result,

$$
\|\nabla^{p+1} d_q(\mathbf{x})\| = \sigma(q-1) \times \cdots \times (q-p) \|\mathbf{x}\|^{q-p-1} \leq C \cdot D^{q-p-1}.
$$

This indicates that $d_q(\mathbf{x})$ is $p^{th}$-order smooth with parameter $C \cdot D^{q-p-1}$, which is equivalent to $\forall\, \mathbf{x}, \mathbf{y} \in \mathcal{Q},$

$$
\|\nabla^p d_q(\mathbf{x}) - \nabla^p d_q(\mathbf{y})\| \leq C \cdot D^{q-p-1} \|\mathbf{x} - \mathbf{y}\|.
$$

Given that $\|\mathbf{x} - \mathbf{y}\| = \|\mathbf{x} - \mathbf{y}\|^{1-\nu}\|\mathbf{x} - \mathbf{y}\|^{\nu} \leq (\|\mathbf{x}\| + \|\mathbf{y}\|)^{1-\nu}\|\mathbf{x} - \mathbf{y}\|^{\nu} \leq (2D)^{1-\nu}\|\mathbf{x} - \mathbf{y}\|^{\nu}$, we have

$$\|\nabla^p d_q(\mathbf{x}) - \nabla^p d_q(\mathbf{y})\| \leq 2^{1-\nu}C \cdot D^{q-p-\nu}\|\mathbf{x} - \mathbf{y}\|^{\nu} \leq \frac{H}{2}\|\mathbf{x} - \mathbf{y}\|^{\nu}.$$

That is, $d_q(\mathbf{x})$ is $p^{th}$-order Hölder smooth with parameter $\frac{H}{2}$ on domain $\mathcal{Q}$. Since $f$ is also $p^{th}$-order Hölder smooth with parameter $\frac{H}{2}$, we conclude that $F = f + d_q$ is $p^{th}$-order Hölder smooth with parameter $H$ on domain $\mathcal{Q}$.

$\square$

**Lemma 5.** *For* $R(\mathbf{x}) = \beta \max_{k\in[T]} \xi_k \langle \mathbf{e}_{\alpha(k)}, \mathbf{x} \rangle + \frac{\sigma}{q}\|\mathbf{x}\|^q$, *we have*

$$R(\mathbf{x}) - \beta(T-1)\delta \leq F(\mathbf{x}) \leq R(\mathbf{x}) + \frac{5}{4}p\beta\rho\sqrt{d}.$$

*Proof.* Since $F(\mathbf{x})$ is constructed with softmax smoothing, we are now able to characterize it with the properties in Lemma 3. $F(\mathbf{x})$ can be upper bounded using the second inequality of Lemma 3 (ii):

$$F(\mathbf{x}) = \beta G_T(\mathbf{x}) + \frac{\sigma}{q}\|\mathbf{x}\|^q$$

$$\leq \beta g_T(\mathbf{x}) + \frac{5}{4}p\beta\rho\sqrt{d} + \frac{\sigma}{q}\|\mathbf{x}\|^q$$

$$= \beta \max_{k\in[T]}\left\{\xi_k \langle \mathbf{e}_{\alpha(k)}, \mathbf{x} \rangle - (k-1)\delta\right\} + \frac{5}{4}p\beta\rho\sqrt{d} + \frac{\sigma}{q}\|\mathbf{x}\|^q$$

$$\leq \beta \max_{k\in[T]} \xi_k \langle \mathbf{e}_{\alpha(k)}, \mathbf{x} \rangle + \frac{5}{4}p\beta\rho\sqrt{d} + \frac{\sigma}{q}\|\mathbf{x}\|^q.$$

$F(\mathbf{x})$ can be lower bounded using the first inequality of Lemma 3 (ii):

$$F(\mathbf{x}) = \beta G_T(\mathbf{x}) + \frac{\sigma}{q}\|x\|^q$$

$$\geq \beta g_T(\mathbf{x}) + \frac{\sigma}{q}\|\mathbf{x}\|^q$$

$$= \beta \max_{k\in[T]}\left\{\xi_k \langle \mathbf{e}_{\alpha(k)}, \mathbf{x} \rangle - (k-1)\delta\right\} + \frac{\sigma}{q}\|\mathbf{x}\|^q$$

$$\geq \beta \max_{k\in[T]} \xi_k \langle \mathbf{e}_{\alpha(k)}, \mathbf{x} \rangle - (T-1)\delta + \frac{\sigma}{q}\|\mathbf{x}\|^q.$$

$\square$

**Lemma 7.** $F(\mathbf{x}_T) - F(\mathbf{x}^*) \geq -\beta(T-1)\delta - \frac{5}{4}p\beta\rho\sqrt{d} + \frac{q-1}{q}\left(\frac{\beta^q}{\sigma T^{\frac{q}{2}}}\right)^{\frac{1}{q-1}}$.

*Proof.*

$$F(\mathbf{x}^*) = \min_{\mathbf{x}} F(\mathbf{x})$$

$$\leq \min_{\mathbf{x}} R(\mathbf{x}) + \frac{5}{4}p\beta\rho\sqrt{d}$$

$$= \min_{\mathbf{x}}\left\{\beta \max_{k\in[T]} \xi_k \langle \mathbf{e}_{\alpha(k)}, \mathbf{x} \rangle + \frac{\sigma}{q}\|\mathbf{x}\|^q\right\} + \frac{5}{4}p\beta\rho\sqrt{d}.$$

Define $\gamma = \left|\max_{k\in[T]} \xi_k \langle \mathbf{e}_{\alpha(k)}, \mathbf{x} \rangle\right|$. Then by symmetry (Doikov, 2022),

$$\|\mathbf{x}\|^q = T^{\frac{q}{2}}\gamma^q.$$

As a result,

$$\min_{\mathbf{x}} R(\mathbf{x}) = \min_{\mathbf{x}}\left\{\beta \max_{k\in[T]} \xi_k \langle \mathbf{e}_{\alpha(k)}, \mathbf{x} \rangle + \frac{\sigma}{q}\|\mathbf{x}\|^q\right\}$$

$$= \min_{\gamma>0}\left\{-\beta\gamma + \frac{\sigma}{q}T^{\frac{q}{2}}\gamma^q\right\}$$

$$= -\frac{q-1}{q}\left(\frac{\beta^q}{\sigma T^{\frac{q}{2}}}\right)^{\frac{1}{q-1}}.$$

Therefore,

$$F(\mathbf{x}^*) \leq -\frac{q-1}{q}\left(\frac{\beta^q}{\sigma T^{\frac{q}{2}}}\right)^{\frac{1}{q-1}} + \frac{5}{4}p\beta\rho\sqrt{d}.$$

Furthermore, for some $\mathbf{x}_T$ generated following some algorithm $\mathcal{A}$ along some trajectory, by definition,

$$g_T(\mathbf{x}_T) \geq \left|\langle e_{\alpha(T)}, \mathbf{x}_T\rangle\right| - (T-1)\delta$$
$$\geq -(T-1)\delta.$$

Therefore,

$$F(\mathbf{x}_T) = f(\mathbf{x}_T) + \frac{\sigma}{q}\|\mathbf{x}_T\|^q$$
$$\geq f(\mathbf{x}_T)$$
$$= \beta G_T(\mathbf{x}_T)$$
$$\geq \beta g_T(\mathbf{x}_T) \qquad\qquad \text{(Lemma 3 (ii))}$$
$$\geq -\beta(T-1)\delta.$$

Given the upper bound on $F(\mathbf{x}^*)$, we have

$$F(\mathbf{x}_T) - F(\mathbf{x}^*) \geq -\beta(T-1)\delta - \frac{5}{4}p\beta\rho\sqrt{d} + \frac{q-1}{q}\left(\frac{\beta^q}{\sigma T^{\frac{q}{2}}}\right)^{\frac{1}{q-1}}.$$

$\square$

# B  PROOF FOR TECHNICAL LEMMAS IN SECTION 5

**Lemma 8.** $\mathbf{x}^* = \arg\min_{\mathbf{x}} f(\mathbf{x})$, $\mathbf{y} = \arg\min_{\mathbf{x}} \tilde{f}(\mathbf{x})$. *(i)* $\forall i \in [\tilde{T}]$, $\langle\mathbf{v}_i, \mathbf{x}^*\rangle = y_i$. *(ii)* $\|\mathbf{x}^*\| = \|\mathbf{y}\|$.

*Proof.* (i) By definition, $f$ is a scaling and rotation of $\tilde{f}$. Since $\mathbf{v}_1, \cdots, \mathbf{v}_{\tilde{T}}$, we can write for $V = [\mathbf{v}_1, \cdots, \mathbf{v}_{\tilde{T}}]$, $f(\mathbf{x}) = \frac{H}{2^{p+\nu+1}(p+\nu-1)!}\tilde{f}(V\mathbf{x})$. Therefore,

$$\mathbf{y} = \arg\min_{\mathbf{x}} \tilde{f}(\mathbf{x})$$
$$= V\arg\min_{\mathbf{x}} \tilde{f}(V\mathbf{x})$$
$$= V\arg\min_{\mathbf{x}} f(\mathbf{x})$$
$$= V\mathbf{x}^*.$$

(ii) This can be shown in the same way as (Arjevani et al., 2019, Lemma 6). $\square$

**Lemma 9.** $f(\mathbf{x})$ *is (i) uniformly convex with degree $q$ and parameter $\sigma$. (ii) $p^{th}$-order Hölder smooth with degree $\nu$ and parameter $H$.*

*Proof.* (i) The proof is similar to that for Lemma 4 (i).

(ii) Without the loss of generality, let the basis that defines $f$ be the standard basis. $\forall\, i \in [\tilde{T}]$, denote $\mathbf{e}_i$ the $i^{th}$ vector in the standard basis. Denote function $g(x) = \frac{1}{p+\nu}|x|^{p+\nu}$. $g^{(p)}(x)$, the $p^{th}$-order derivative of $g(x)$ is $(p+\nu-1)!x^\nu$ if $p$ is odd, $(p+\nu-1)!|x|^\nu$ is even. Let $\mathbf{d}_i = \mathbf{e}_i - \mathbf{e}_{i+1}$, then

$$f(\mathbf{x}) = \frac{H}{2^{p+\nu+1}(p+\nu-1)!}\left(\frac{1}{p+\nu}\sum_{i=1}^{\tilde{T}} g(\langle\mathbf{d}_i, \mathbf{x}\rangle) - \gamma x_1\right) + \frac{\sigma}{q}\|\mathbf{x}\|^q$$

Since $q < p + \nu$, then $q \leq p$. Therefore, $\forall\, \mathbf{x}, \mathbf{y} \in \mathbb{R}^d$,

$$
\begin{aligned}
\|\nabla^p f(\mathbf{x}) - \nabla^p f(\mathbf{y})\| &= \frac{H}{2^{p+\nu+1}(p+\nu-1)!} \Big\| \sum_{i=1}^{\tilde{T}} \Big[ g^{(p)}(\langle \mathbf{d}_i, \mathbf{x} \rangle) - g^{(p)}(\langle \mathbf{d}_i, \mathbf{y} \rangle) \Big] [\mathbf{d}_i]^p \Big\| \\
&\leq \frac{H(p+\nu-1)!}{2^{p+\nu+1}(p+\nu-1)!} \Big\| \sum_{i=1}^{\tilde{T}} |\langle \mathbf{d}_i, \mathbf{x} - \mathbf{y} \rangle|^{\nu} [\mathbf{d}_i]^p \Big\| \\
&\leq \frac{H}{2^{p+\nu+1}} \sqrt{2} \|\mathbf{x} - \mathbf{y}\|^{\nu} \Big\| \sum_{i=1}^{\tilde{T}} [\mathbf{d}_i]^p \Big\| \\
&\leq \frac{H}{2^{p+\nu+1}} \sqrt{2} \|\mathbf{x} - \mathbf{y}\|^{\nu} 2^p \\
&\leq H \|\mathbf{x} - \mathbf{y}\|^{\nu}.
\end{aligned}
$$

$\square$

**Lemma 15.** *For* $\mathbf{y} = \arg\min_{\mathbf{x}} \tilde{f}(\mathbf{x})$,

(i) $y_1 \geq y_2 \geq \cdots \geq y_{\tilde{T}} \geq 0$.

(ii) $y_{t+1} = y_t - \left( \gamma - \tilde{\sigma} \|\mathbf{y}\|^{q-2} \sum_{j=1}^{t} y_j \right)^{\frac{1}{p+\nu-1}}$.

(iii) $\sum_{i=1}^{\tilde{T}} y_i = \frac{\gamma}{\tilde{\sigma} \|\mathbf{y}\|^{q-2}}$.

*Proof.* (i) The proof is similar to (Arjevani et al., 2019, Lemma 1), relying on the fact that $\tilde{f}$ is strictly convex, which holds true for our higher-order construction as well, since the function is uniformly convex.

(ii)

$$
\nabla \tilde{f}(y) = \begin{bmatrix}
|y_1 - y_2|^{p+\nu-2} (y_1 - y_2) - \gamma + \tilde{\sigma} \|\mathbf{y}\|^{q-2} y_1 \\
|y_2 - y_1|^{p+\nu-2} (y_2 - y_1) + |y_2 - y_3|^{p+\nu-2} (y_2 - y_3) + \tilde{\sigma} \|\mathbf{y}\|^{q-2} y_2 \\
\vdots \\
|y_{\tilde{T}-1} - y_{\tilde{T}-2}|^{p+\nu-2} (y_{\tilde{T}-1} - y_{\tilde{T}-2}) + |y_{\tilde{T}-1} - y_{\tilde{T}}|^{p+\nu-2} (y_{\tilde{T}-1} - y_{\tilde{T}}) + \tilde{\sigma} \|\mathbf{y}\|^{q-2} y_{\tilde{T}-1} \\
|y_{\tilde{T}} - y_{\tilde{T}-1}|^{p+\nu-2} (y_{\tilde{T}} - y_{\tilde{T}-1}) + \tilde{\sigma} \|\mathbf{y}\|^{q-2} y_{\tilde{T}}
\end{bmatrix}
$$

Given that $y_1 \geq y_2 \geq \cdots \geq y_{\tilde{T}} \geq 0$, we have $\forall i \in [\tilde{T} - 1]$, $|y_i - y_{i+1}| = y_i - y_{i+1}$. Therefore, with $\nabla \tilde{f}(y) = 0$, we have

$$
(y_1 - y_2)^{p+\nu-1} = \gamma - \tilde{\sigma} \|\mathbf{y}\|^{q-2} y_1, \tag{2}
$$

$$
(y_{i-1} - y_i)^{p+\nu-1} = (y_i - y_{i+1})^{p+\nu-1} + \tilde{\sigma} \|\mathbf{y}\|^{q-2} y_i, \quad 2 \leq i \leq \tilde{T} - 1, \tag{3}
$$

$$
(y_{\tilde{T}-1} - y_{\tilde{T}})^{p+\nu-1} = \tilde{\sigma} \|\mathbf{y}\|^{q-2} y_{\tilde{T}}. \tag{4}
$$

Summing Eq. (2) and (3), we have

$$
(y_i - y_{i+1})^{p+\nu-1} = \gamma - \tilde{\sigma} \|\mathbf{y}\|^{q-2} \sum_{j=1}^{i} y_j,
$$

which completes the proof.

(iii) We know from part (ii) that

$$
(y_{\tilde{T}-1} - y_{\tilde{T}})^{p+\nu-1} = \gamma - \tilde{\sigma} \|\mathbf{y}\|^{q-2} \sum_{j=1}^{\tilde{T}-1} y_j.
$$

Grouping this with Eq. (4), we have $\tilde{\sigma} \|\mathbf{y}\|^{q-2} y_{\tilde{T}} = \gamma - \tilde{\sigma} \|\mathbf{y}\|^{q-2} \sum_{j=1}^{\tilde{T}-1} y_j$, which yields the desired result. $\square$

**Lemma 11.** $\|\mathbf{y}\| \leq \dfrac{2^{\frac{2}{3q-2}}\gamma^{\frac{3(p+\nu)-2}{(p+\nu-1)(3q-2)}}}{\tilde{\sigma}^{\frac{3}{3q-2}}}.$

*Proof.* By Lemma 15 (iii) and Lemma 10 (ii),

$$\|\mathbf{y}\|_2^2 \leq \|\mathbf{y}\|_1 \|\mathbf{y}\|_\infty$$

$$= \max_{i\in[\tilde{T}]} |y_i| \times \sum_{i=1}^{d} y_i$$

$$= y_1 \times \sum_{i=1}^{\tilde{T}} y_i$$

$$\leq \left(\gamma^{\frac{1}{p+\nu-1}} + \sqrt{\frac{2\gamma^{\frac{p+\nu}{p+\nu-1}}}{\tilde{\sigma}\|\mathbf{y}\|^{q-2}}}\right) \times \frac{\gamma}{\tilde{\sigma}\|\mathbf{y}\|^{q-2}}$$

$$= \left(1 + \sqrt{\frac{2\gamma^{\frac{p+\nu-2}{p+\nu-1}}}{\tilde{\sigma}\|\mathbf{y}\|^{q-2}}}\right) \frac{\gamma^{\frac{p+\nu}{p+\nu-1}}}{\tilde{\sigma}\|\mathbf{y}\|^{q-2}}$$

Let $\gamma \geq \left(3\tilde{\sigma}\|\mathbf{y}\|^{q-2}\right)^{\frac{p+\nu-1}{p+\nu-2}}$, then we have $\frac{\gamma^{\frac{p+\nu-2}{p+\nu-1}}}{3\tilde{\sigma}\|\mathbf{y}\|^{q-2}} \geq 1$ and moreover $\sqrt{\frac{\gamma^{\frac{p+\nu-2}{p+\nu-1}}}{3\tilde{\sigma}\|\mathbf{y}\|^{q-2}}} \geq 1$, so that we can merge the terms as follows:

$$\|\mathbf{y}\|^2 \leq \left(\sqrt{\frac{\gamma^{\frac{p+\nu-2}{p+\nu-1}}}{3\tilde{\sigma}\|\mathbf{y}\|^{q-2}}} + \sqrt{\frac{2\gamma^{\frac{p+\nu-2}{p+\nu-1}}}{\tilde{\sigma}\|\mathbf{y}\|^{q-2}}}\right) \frac{\gamma^{\frac{p+\nu}{p+\nu-1}}}{\tilde{\sigma}\|\mathbf{y}\|^{q-2}}$$

$$= \left(\sqrt{\frac{1}{3}} + \sqrt{2}\right) \frac{\gamma^{\frac{p+\nu-2}{2(p+\nu-1)} + \frac{p+\nu}{p+\nu-1}}}{\tilde{\sigma}^{\frac{3}{2}}\|\mathbf{y}\|^{\frac{3(q-2)}{2}}}$$

$$\leq \frac{2\gamma^{\frac{3(p+\nu)-2}{2(p+\nu-1)}}}{\tilde{\sigma}^{\frac{3}{2}}\|\mathbf{y}\|^{\frac{3(q-2)}{2}}}.$$

We can solve for $\|\mathbf{y}\| \leq \left(\dfrac{2\gamma^{\frac{3(p+\nu)-2}{2(p+\nu-1)}}}{\tilde{\sigma}^{\frac{3}{2}}}\right)^{\frac{2}{3q-2}} = \dfrac{2^{\frac{2}{3q-2}}\gamma^{\frac{3(p+\nu)-2}{(p+\nu-1)(3q-2)}}}{\tilde{\sigma}^{\frac{3}{3q-2}}}.$ $\qquad\square$

**Lemma 10.** *For $\mathbf{y} = \arg\min_{\mathbf{x}} \tilde{f}(\mathbf{x})$,*

*(i)* $\forall t \in [\tilde{T}]$, $y_t \geq y_1 - (t-1)\gamma^{\frac{1}{p+\nu-1}}$.

*(ii)* *For* $\tilde{T} = \left\lfloor \frac{y_1}{\gamma^{\frac{1}{p+\nu-1}}} + 1 \right\rfloor$, $y_1 \leq \gamma^{\frac{1}{p+\nu-1}} + \sqrt{\frac{2\gamma^{\frac{p+\nu}{p+\nu-1}}}{\tilde{\sigma}\|\mathbf{y}\|^{q-2}}}$.

*(iii)* *For* $\gamma \geq \tilde{\sigma}^{\frac{p+\nu-1}{p+\nu-2}}\|\mathbf{y}\|^{\frac{(p+\nu-1)(q-2)}{p+\nu-2}}$, $\forall t \in [\tilde{T}]$, $y_t \geq \frac{\gamma^{\frac{p+\nu}{2(p+\nu-1)}}}{2^{p+\nu+1}\tilde{\sigma}^{\frac{1}{2}}\|\mathbf{y}\|^{\frac{q-2}{2}}} + \left(\frac{1}{2} - i\right)\gamma^{\frac{1}{p+\nu-1}}$.

*Proof.* (i) By Lemma 15 (ii), $\forall i \in [\tilde{T}]$

$$y_i = y_{i-1} - \left(\gamma - \tilde{\sigma}\|\mathbf{y}\|^{q-2}\sum_{j=1}^{i-1} y_j\right)^{\frac{1}{p+\nu-1}}$$

$$\geq y_{i-1} - \gamma^{\frac{1}{p+\nu-1}}$$

$$\geq y_1 - (i-1)\gamma^{\frac{1}{p+\nu-1}},$$

in which the first inequality follows from Lemma 15 (i) that $\forall i \in [\tilde{T}]$, $y_i \geq 0$, and the second inequality follows from applying the first inequality recursively.

(ii) It follows from part (i) that

$$\sum_{i=1}^{\tilde{T}} y_i \geq \sum_{i=1}^{\tilde{T}} \max\left\{0, y_1 - (i-1)\gamma^{\frac{1}{p+\nu-1}}\right\}.$$

For $\tilde{T} = \left\lfloor \frac{y_1}{\gamma^{\frac{1}{p+\nu-1}}} + 1 \right\rfloor \leq \frac{y_1}{\gamma^{\frac{1}{p+\nu-1}}} + 1$, we always have $y_1 - (\tilde{T}-1)\gamma^{\frac{1}{p+\nu-1}} \geq 0$. Consequently, $\forall i \in [\tilde{T}], y_1 - (i-1)\gamma^{\frac{1}{p+\nu-1}} \geq 0$. Therefore,

$$\sum_{i=1}^{\tilde{T}} y_i \geq \sum_{i=1}^{\tilde{T}} y_1 - (i-1)\gamma^{\frac{1}{p+\nu-1}}$$

$$= \sum_{i=1}^{\left\lfloor y_1/\gamma^{\frac{1}{p+\nu-1}}+1 \right\rfloor} y_1 - (i-1)\gamma^{\frac{1}{p+\nu-1}}$$

$$= \left\lfloor \frac{y_1}{\gamma^{\frac{1}{p+\nu-1}}} + 1 \right\rfloor \cdot y_1 - \gamma^{\frac{1}{p+\nu-1}} \cdot \frac{\left\lfloor \frac{y_1}{\gamma^{\frac{1}{p+\nu-1}}} + 1 \right\rfloor \left( \left\lfloor \frac{y_1}{\gamma^{\frac{1}{p+\nu-1}}} + 1 \right\rfloor - 1 \right)}{2}$$

$$\geq \frac{y_1}{\gamma^{\frac{1}{p+\nu-1}}} \cdot y_1 - \gamma^{\frac{1}{p+\nu-1}} \cdot \frac{\left( \frac{y_1}{\gamma^{\frac{1}{p+\nu-1}}} + 1 \right) \left( \frac{y_1}{\gamma^{\frac{1}{p+\nu-1}}} + 1 - 1 \right)}{2}$$

$$= \frac{y_1^2}{\gamma^{\frac{1}{p+\nu-1}}} - \frac{y_1^2}{2\gamma^{\frac{1}{p+\nu-1}}} - \frac{y_1}{2}$$

$$= \frac{y_1}{2}\left( \frac{y_1}{\gamma^{\frac{1}{p+\nu-1}}} - 1 \right).$$

Combining with Lemma 15 (iii) that $\sum_{i=1}^{\tilde{T}} y_i = \frac{\gamma}{\tilde{\sigma}\|\mathbf{y}\|^{q-2}}$, we have

$$\frac{\gamma}{\tilde{\sigma}\|\mathbf{y}\|^{q-2}} = \sum_{i=1}^{\tilde{T}} y_i \geq \frac{y_1}{2}\left( \frac{y_1}{\gamma^{\frac{1}{p+\nu-1}}} - 1 \right).$$

Equivalently,

$$y_1^2 - \gamma^{\frac{1}{p+\nu-1}} y_1 - \frac{2\gamma^{\frac{p+\nu}{p+\nu-1}}}{\tilde{\sigma}\|\mathbf{y}\|^{q-2}} \leq 0.$$

By the quadratic formula, we have

$$y_1 \leq \frac{\gamma^{\frac{1}{p+\nu-1}} + \sqrt{\gamma^{\frac{2}{p+\nu-1}} + \frac{8\gamma^{\frac{p+\nu}{p+\nu-1}}}{\tilde{\sigma}\|\mathbf{y}\|^{q-2}}}}{2}$$

$$\leq \frac{\gamma^{\frac{1}{p+\nu-1}} + \sqrt{\gamma^{\frac{2}{p+\nu-1}}} + \sqrt{\frac{8\gamma^{\frac{p+\nu}{p+\nu-1}}}{\tilde{\sigma}\|\mathbf{y}\|^{q-2}}}}{2}$$

$$= \gamma^{\frac{1}{p+\nu-1}} + \sqrt{\frac{2\gamma^{\frac{p+\nu}{p+\nu-1}}}{\tilde{\sigma}\|\mathbf{y}\|^{q-2}}}$$

(iii) Since $\sum_{i=1}^{\tilde{T}} y_i = \frac{\gamma}{\tilde{\sigma}\|\mathbf{y}\|^{q-2}}$, $\exists\, t_0 \in [\tilde{T}]$ such that $\sum_{i=1}^{t_0} y_i > (1 - \frac{1}{2^{p+\nu-1}})\frac{\gamma}{\tilde{\sigma}\|\mathbf{y}\|^{q-2}}$ and $\forall t < t_0$, $\sum_{i=1}^{t} y_i \leq (1 - \frac{1}{2^{p+\nu-1}})\frac{\gamma}{\tilde{\sigma}\|\mathbf{y}\|^{q-2}}$. Then $\forall\, i < t_0$, we can merge the terms in Lemma 15 (ii) as

follows:

$$y_{i+1} = y_i - \left(\gamma - \tilde{\sigma}\|\mathbf{y}\|^{q-2}\sum_{j=1}^{i}y_j\right)^{\frac{1}{p+\nu-1}}$$

$$\leq y_i - \left(\gamma - (1 - \frac{1}{2^{p+\nu-1}})\gamma\right)^{\frac{1}{p+\nu-1}}$$

$$= y_i - \frac{\gamma^{\frac{1}{p+\nu-1}}}{2}.$$

Applying this relation recursively, we have

$$y_{t_0} \leq y_{t_0-1} - \frac{\gamma^{\frac{1}{p+\nu-1}}}{2} \leq \cdots \leq y_1 - (t_0-1)\frac{\gamma^{\frac{1}{p+\nu-1}}}{2}.$$

Given that $y_{t_0} \geq 0$, this yields $y_1 \geq (t_0 - 1)\frac{\gamma^{\frac{1}{p+\nu-1}}}{2}$.

Now we characterize $t_0$. By definition, we have $\sum_{i=1}^{t_0} y_i > (1 - \frac{1}{2^{p+\nu-1}})\frac{\gamma}{\tilde{\sigma}\|\mathbf{y}\|^{q-2}}$. In the meantime,

$$\sum_{i=1}^{t_0} y_i \leq \sum_{i=1}^{t_0} y_1$$

$$= t_0 y_1$$

$$\leq t_0 \left(\gamma^{\frac{1}{p+\nu-1}} + \sqrt{\frac{2\gamma^{\frac{p+\nu}{p+\nu-1}}}{\tilde{\sigma}\|\mathbf{y}\|^{q-2}}}\right),$$

where the first inequality follows from Lemma 15 (i) and the second from part (ii). Together, we have $(1 - \frac{1}{2^{p+\nu-1}})\frac{\gamma}{\tilde{\sigma}\|\mathbf{y}\|^{q-2}} < t_0 \left(\gamma^{\frac{1}{p+\nu-1}} + \sqrt{\frac{2\gamma^{\frac{p+\nu}{p+\nu-1}}}{\tilde{\sigma}\|\mathbf{y}\|^{q-2}}}\right)$, from which we solve for

$$t_0 > \frac{(2^{p+\nu-1} - 1)\gamma^{\frac{p+\nu-2}{p+\nu-1}}}{2^{p+\nu-1}\left(\tilde{\sigma}\|\mathbf{y}\|^{q-2} + \sqrt{2\tilde{\sigma}\gamma^{\frac{p+\nu-2}{p+\nu-1}}\|\mathbf{y}\|^{q-2}}\right)}$$

Plugging this characterization of $t_0$ back in,

$$y_1 \geq (t_0 - 1)\frac{\gamma^{\frac{1}{p+\nu-1}}}{2}$$

$$> \left(\frac{(2^{p+\nu-1} - 1)\gamma^{\frac{p+\nu-2}{p+\nu-1}}}{2^{p+\nu-1}\left(\tilde{\sigma}\|\mathbf{y}\|^{q-2} + \sqrt{2\tilde{\sigma}\gamma^{\frac{p+\nu-2}{p+\nu-1}}\|\mathbf{y}\|^{q-2}}\right)} - 1\right)\frac{\gamma^{\frac{1}{p+\nu-1}}}{2}$$

$$= \frac{(2^{p+\nu-1} - 1)\gamma}{2^{p+\nu+1}\left(\tilde{\sigma}\|\mathbf{y}\|^{q-2} + \sqrt{2\tilde{\sigma}\gamma^{\frac{p+\nu-2}{p+\nu-1}}\|\mathbf{y}\|^{q-2}}\right)} - \frac{\gamma^{\frac{1}{p+\nu-1}}}{2}.$$

Finally, plugging this into the result from part (i), $\forall i \in [\tilde{T}]$,

$$y_i \geq y_1 - (i-1)\gamma^{\frac{1}{p+\nu-1}}$$

$$\geq \frac{(2^{p+\nu-1} - 1)\gamma}{2^{p+\nu+1}\left(\tilde{\sigma}\|\mathbf{y}\|^{q-2} + \sqrt{2\tilde{\sigma}\gamma^{\frac{p+\nu-2}{p+\nu-1}}\|\mathbf{y}\|^{q-2}}\right)} - \frac{\gamma^{\frac{1}{p+\nu-1}}}{2} - (i-1)\gamma^{\frac{1}{p+\nu-1}}$$

$$= \frac{(2^{p+\nu-1} - 1)\gamma}{2^{p+\nu+1}\left(\tilde{\sigma}\|\mathbf{y}\|^{q-2} + \sqrt{2\tilde{\sigma}\gamma^{\frac{p+\nu-2}{p+\nu-1}}\|\mathbf{y}\|^{q-2}}\right)} + \left(\frac{1}{2} - i\right)\gamma^{\frac{1}{p+\nu-1}}$$

By letting $\gamma \geq \tilde{\sigma}^{\frac{p+\nu-1}{p+\nu-2}}\|\mathbf{y}\|^{\frac{(p+\nu-1)(q-2)}{p+\nu-2}}$ as stated in the condition, we have $\gamma^{\frac{p+\nu-2}{p+\nu-1}} \geq \tilde{\sigma}\|\mathbf{y}\|^{q-2}$ and are able to merge the terms as follows:

$$y_i \geq \frac{(2^{p+\nu-1}-1)\gamma}{2^{p+\nu+1}\left((\tilde{\sigma}\|\mathbf{y}\|^{q-2})^{\frac{1}{2}}\cdot(\tilde{\sigma}\|\mathbf{y}\|^{q-2})^{\frac{1}{2}}+\sqrt{2}\,(\tilde{\sigma}\|\mathbf{y}\|^{q-2})^{\frac{1}{2}}\cdot\left(\gamma^{\frac{p+\nu-2}{p+\nu-1}}\right)^{\frac{1}{2}}\right)}+\left(\frac{1}{2}-i\right)\gamma^{\frac{1}{p+\nu-1}}$$

$$\geq \frac{(2^{p+\nu-1}-1)\gamma}{2^{p+\nu+1}\left((\tilde{\sigma}\|\mathbf{y}\|^{q-2})^{\frac{1}{2}}\cdot\left(\gamma^{\frac{p+\nu-2}{p+\nu-1}}\right)^{\frac{1}{2}}+\sqrt{2}\,(\tilde{\sigma}\|\mathbf{y}\|^{q-2})^{\frac{1}{2}}\cdot\left(\gamma^{\frac{p+\nu-2}{p+\nu-1}}\right)^{\frac{1}{2}}\right)}+\left(\frac{1}{2}-i\right)\gamma^{\frac{1}{p+\nu-1}}$$

$$\geq \frac{(2^{p+\nu-1}-1)\gamma}{2^{p+\nu+1}\left((1+\sqrt{2})(\tilde{\sigma}\|\mathbf{y}\|^{q-2})^{\frac{1}{2}}\cdot\left(\gamma^{\frac{p+\nu-2}{p+\nu-1}}\right)^{\frac{1}{2}}\right)}+\left(\frac{1}{2}-i\right)\gamma^{\frac{1}{p+\nu-1}}$$

$$\geq \frac{\gamma^{\frac{p+\nu}{2(p+\nu-1)}}}{2^{p+\nu+1}\tilde{\sigma}^{\frac{1}{2}}\|\mathbf{y}\|^{\frac{q-2}{2}}}+\left(\frac{1}{2}-i\right)\gamma^{\frac{1}{p+\nu-1}}$$

$\square$

**Lemma 12.** *For* $\mathbf{y} = \arg\min_{\mathbf{x}} \tilde{f}(\mathbf{x})$, *let* $t_1 \in [\tilde{T}]$ *be such that* $y_{t_1} > \frac{1}{p+\nu-1}\tilde{\sigma}^{\frac{1}{p+\nu-2}}\|\mathbf{y}\|^{\frac{q-2}{p+\nu-2}}$ *and* $y_{t_1+1} \leq \frac{1}{p+\nu-1}\tilde{\sigma}^{\frac{1}{p+\nu-2}}\|\mathbf{y}\|^{\frac{q-2}{p+\nu-2}}$. *Then*

*(i)* $\forall i \in [\tilde{T}]$, $y_i = y_{i+1} + \left(\tilde{\sigma}\|\mathbf{y}\|^{q-2}\sum_{j=i+1}^{\tilde{T}} y_j\right)^{\frac{1}{p+\nu-1}}$ *and* $y_{i+1} \leq \frac{1}{\tilde{\sigma}\|\mathbf{y}\|^{q-2}}y_i^p$

*(ii)* $\forall i \geq t_1$, $\left(\frac{1}{c_{p,\nu}}\right)^{p+\nu-1}\frac{1}{\tilde{\sigma}\|\mathbf{y}\|^{q-2}}y_i^{p+\nu-1} \leq y_{i+1}$ *where* $c_{p,\nu}$ *is a constant depending on* $p$, $\nu$.

*(iii)* $\forall i \leq \tilde{T} - t_1$, $y_{t_1+i} \geq \left(\frac{1}{c_{p,\nu}}\right)^{\frac{(p+\nu-1)((p+\nu-1)^i-1)}{p+\nu-2}}\left(\tilde{\sigma}\|\mathbf{y}\|^{q-2}\right)^{\frac{1}{p+\nu-2}}(p+\nu-1)^{-(p+\nu-1)^i}$.

*Proof.* (i) Starting from Lemma 15 (ii), $\forall i \in [\tilde{T}]$,

$$y_i = y_{i+1} + \left(\gamma - \tilde{\sigma}\|\mathbf{y}\|^{q-2}\sum_{j=1}^{i} y_j\right)^{\frac{1}{p+\nu-1}}$$

$$= y_{i+1} + \left(\gamma - \tilde{\sigma}\|\mathbf{y}\|^{q-2}\sum_{j=1}^{\tilde{T}} y_j + \tilde{\sigma}\|\mathbf{y}\|^{q-2}\sum_{j=i+1}^{\tilde{T}} y_j\right)^{\frac{1}{p+\nu-1}}$$

$$= y_{i+1} + \left(\gamma - \tilde{\sigma}\|\mathbf{y}\|^{q-2}\cdot\frac{\gamma}{\tilde{\sigma}\|\mathbf{y}\|^{q-2}} + \tilde{\sigma}\|\mathbf{y}\|^{q-2}\sum_{j=i+1}^{\tilde{T}} y_j\right)^{\frac{1}{p+\nu-1}} \qquad \text{(Lemma 15 (iii))}$$

$$= y_{i+1} + \left(\tilde{\sigma}\|\mathbf{y}\|^{q-2}\sum_{j=i+1}^{\tilde{T}} y_j\right)^{\frac{1}{p+\nu-1}}.$$

Since $x_{i+1} \geq 0$, we have

$$y_i \geq \left(\tilde{\sigma}\|\mathbf{y}\|^{q-2}\sum_{j=i+1}^{\tilde{T}} y_j\right)^{\frac{1}{p+\nu-1}}$$

$$\geq \left(\tilde{\sigma}\|\mathbf{y}\|^{q-2}y_{i+1}\right)^{\frac{1}{p+\nu-1}},$$

equivalently,

$$y_{i+1} \leq \frac{1}{\tilde{\sigma}\|\mathbf{y}\|^{q-2}}y_i^{p+\nu-1}.$$

(ii)

$$\sum_{j=i+1}^{\tilde{T}} y_j = y_{i+1} + y_{i+2} + \cdots + y_{\tilde{T}}$$

$$\leq y_{i+1} + \frac{1}{\tilde{\sigma}\|\mathbf{y}\|^{q-2}} y_{i+1}^{p+\nu-1} + \frac{1}{(\tilde{\sigma}\|\mathbf{y}\|^{q-2})^{(p+\nu-1)+1}} y_{i+1}^{(p+\nu-1)^2} + \cdots$$

$$+ \frac{1}{(\tilde{\sigma}\|\mathbf{y}\|^{q-2})^{\sum_{j=0}^{\tilde{T}-i-2}(p+\nu-1)^j}} y_{j+1}^{(p+\nu-1)^{\tilde{T}-i-1}}$$

$$= y_{i+1} \sum_{j=0}^{\tilde{T}-i-1} \left( \frac{y_{i+1}}{\tilde{\sigma}^{\frac{1}{p+\nu-2}}\|\mathbf{y}\|^{\frac{q-2}{p+\nu-2}}} \right)^{(p+\nu-1)^j - 1}.$$

Given that $i \geq t_1$, then $i + 1 \geq t_1 + 1$ and by Lemma 15 (i) and part (ii) of this lemma, $y_{i+1} \leq y_{t_1+1} \leq \frac{1}{p+\nu-1}\tilde{\sigma}^{\frac{1}{p+\nu-2}}\|\mathbf{y}\|^{\frac{q-2}{p+\nu-2}}$. Therefore,

$$\sum_{j=i+1}^{\tilde{T}} y_j \leq y_{i+1} \sum_{j=0}^{\tilde{T}-i-1} \left( \frac{\frac{1}{p+\nu-1}\tilde{\sigma}^{\frac{1}{p+\nu-2}}\|\mathbf{y}\|^{\frac{q-2}{p+\nu-2}}}{\tilde{\sigma}^{\frac{1}{p+\nu-2}}\|\mathbf{y}\|^{\frac{q-2}{p+\nu-2}}} \right)^{(p+\nu-1)^j - 1}$$

$$= (p+\nu-1)y_{i+1} \sum_{j=0}^{\tilde{T}-i-1} \frac{1}{(p+\nu-1)^{(p+\nu-1)^j}}$$

$$\leq \frac{p+\nu-1}{(p+\nu-2)^2} y_{i+1}$$

With this, we go back to part (i), for $y_{i+1} \leq \frac{1}{p+\nu-1}\tilde{\sigma}^{\frac{1}{p+\nu-2}}\|\mathbf{y}\|^{\frac{q-2}{p+\nu-2}}$,

$$y_i = y_{i+1} + \left( \tilde{\sigma}\|\mathbf{y}\|^{q-2} \sum_{j=i+1}^{\tilde{T}} y_j \right)^{\frac{1}{p+\nu-1}}$$

$$\leq y_{i+1} + \left( \frac{p+\nu-1}{(p+\nu-2)^2}\tilde{\sigma}\|\mathbf{y}\|^{q-2}y_{i+1} \right)^{\frac{1}{p+\nu-1}}$$

$$= y_{i+1}^{\frac{1}{p+\nu-1}} y_{i+1}^{\frac{p+\nu-2}{p+\nu-1}} + \left( \frac{p+\nu-1}{(p+\nu-2)^2}\tilde{\sigma}\|\mathbf{y}\|^{q-2}y_{i+1} \right)^{\frac{1}{p+\nu-1}}$$

$$\leq y_{i+1}^{\frac{1}{p+\nu-1}} \left( \frac{1}{p+\nu-1}\tilde{\sigma}^{\frac{1}{p+\nu-2}}\|\mathbf{y}\|^{\frac{q-2}{p+\nu-2}} \right)^{\frac{p+\nu-2}{p+\nu-1}} + \left( \frac{p+\nu-1}{(p+\nu-2)^2}\tilde{\sigma}\|\mathbf{y}\|^{q-2}y_{i+1} \right)^{\frac{1}{p+\nu-1}}$$

$$= \left( ((p+\nu-1)^{\frac{2-p-\nu}{p+\nu-1}} + \left( \frac{p+\nu-1}{(p+\nu-2)^2} \right)^{\frac{1}{p+\nu-1}} \right) \left( \tilde{\sigma}\|\mathbf{y}\|^{q-2}y_{i+1} \right)^{\frac{1}{p+\nu-1}}$$

$$= c_{p,\nu} \left( \tilde{\sigma}\|\mathbf{y}\|^{q-2}y_{i+1} \right)^{\frac{1}{p+\nu-1}}$$

for $c_{p,\nu} = ((p+\nu-1)^{\frac{2-p-\nu}{p+\nu-1}} + \left( \frac{p+\nu-1}{(p+\nu-2)^2} \right)^{\frac{1}{p+\nu-1}}$. Therefore, $y_{i+1} \geq \left( \frac{1}{c_{p,\nu}} \right)^{p+\nu-1} \frac{1}{\tilde{\sigma}\|\mathbf{y}\|^{q-2}} y_i^{p+\nu-1}$.

(iii) $\forall i \leq \tilde{T} - t_1, t_1 + i \geq t_1$, therefore, applying part (ii) recursively yields

$$
\begin{aligned}
y_{t_1+i} &\geq \left(\frac{1}{c_{p,\nu}}\right)^{p+\nu-1} \frac{(y_{t_1+i-1})^{p+\nu-1}}{\tilde{\sigma}\|\mathbf{y}\|^{q-2}} \\
&\geq \left(\frac{1}{c_{p,\nu}}\right)^{(p+\nu-1)^2+(p+\nu-1)} \frac{(y_{t_1+i-2})^{(p+\nu-1)^2}}{(\tilde{\sigma}\|\mathbf{y}\|^{q-2})^{p+\nu}} \\
&\geq \cdots \\
&\geq \left(\frac{1}{c_{p,\nu}}\right)^{\frac{(p+\nu-1)((p+\nu-1)^i-1)}{p+\nu-2}} \frac{y_{t_1}^{(p+\nu-1)^i}}{(\tilde{\sigma}\|\mathbf{y}\|^{q-2})^{\frac{(p+\nu-1)^i-1}{p+\nu-2}}}
\end{aligned}
$$

By the definition of $t_1$, we know that $y_{t_1} > \frac{1}{p+\nu-1}\tilde{\sigma}^{\frac{1}{p+\nu-2}}\|\mathbf{y}\|^{\frac{q-2}{p+\nu-2}}$. Thus

$$
\begin{aligned}
y_{t_1+i} &\geq \left(\frac{1}{c_{p,\nu}}\right)^{\frac{(p+\nu-1)((p+\nu-1)^i-1)}{(p+\nu-1)-1}} \frac{\left(\frac{1}{p+\nu-1}\tilde{\sigma}^{\frac{1}{p+\nu-2}}\|\mathbf{y}\|^{\frac{q-2}{p+\nu-2}}\right)^{(p+\nu-1)^i}}{(\tilde{\sigma}\|\mathbf{y}\|^{q-2})^{\frac{(p+\nu-1)^i-1}{p+\nu-2}}} \\
&= \left(\frac{1}{c_{p,\nu}}\right)^{\frac{(p+\nu-1)((p+\nu-1)^i-1)}{p+\nu-2}} \left(\tilde{\sigma}\|\mathbf{y}\|^{q-2}\right)^{\frac{1}{p+\nu-2}} (p+\nu-1)^{-(p+\nu-1)^i}
\end{aligned}
$$

$\square$

