# OpenReview forum: "Tight Lower Bounds under Asymmetric High-Order Hölder Smoothness and Uniform Convexity"
_ICLR.cc/2025/Conference — ICLR 2025 Oral_

### Official Review · Reviewer_srAx · 2024-10-22

**Soundness:** 3
**Presentation:** 3
**Contribution:** 3
**Rating:** 8
**Confidence:** 3

**Summary:**

This paper provides general lower bounds for functions, which are uniformly convex with degree $q$ and parameter $\sigma$ and whose $p^{th}$-order derivatives are H&ouml;lder smooth for degree $v$. The proposed lower bounds focus on two asymmetric cases: (1) $q>p+v$ and (2) $q<p+v$, matching the corresponding upper bounds in these cases.

**Strengths:**

* The proposed lower bounds on this work match the corresponding upper bounds for general algorithms in these settings, therefore the results are of great interest to the community.
* The use of the truncated Gaussian Smoothing using the $L_{&infin;}$-norm is an interesting technical novelty to circumvent the dimensionality dependencies that would arise otherwise when using Smoothing.

**Weaknesses:**

The main issue for me that causes confusion in section 4, concerning case (1) $q>p+v$ is the following:

* In line 327 in the lower bound for $F(x_T)-F(x^*)$ the first term is $-2p\beta\rho(T-1+5/8\sqrt{d})$, then you use the assumption that $2T\geq T-1+5/8\sqrt{d}$, however in line 328 your term becomes $-4\beta\rho T$, omitting $p$. I was wondering if there was a reason for this, as it is not discussed and in this setting $p\geq 1$, therefore for me it is not trivial to remove it.

In several places in the paper there are potentially typos that cause me confusion both in the main part and in the appendix. More specifically:

__Line 783__ $\rightarrow$ you say $1/\sqrt{d}$ is a truncated Gaussian, I think you meant to say $Z_i/\sqrt{d}$

__Line 230__ $\rightarrow$ I think it should be Lemma 1, not Lemma 14 according to the derivation in the appendix as well.

__Line 934__ $\rightarrow$ You have a $\beta^v$ appear in the first inequality, which is confusing to me does it not directly arise from the second inequality, where you use Lipschitzness?

__Line 293__ $\rightarrow$ I believe here that $x_t$ should be $x_s$.

__Line 296__ $\rightarrow$ Should the equality here be $\geq$, since the way that $e_{\alpha(s)}$ is chosen is for it to be the maximum possible that has not already been chosen.

__Line 317__ $\rightarrow$ Lemma 14 should be Lemma 1 here or am I mistaken?

__Line 1011__ $\rightarrow$ Similar to 296 it should be $\geq$ or am I mistaken here again?

__Line 1019__ $\rightarrow$ I think it would be beneficial to reference Lemma 1 here for justification as the write up is extremely lengthy and it caused some confusion for me here.

For Section 5, I only have one question. Concerning lines 402,403, what do you mean by scaling of $\tilde{f}$ and how does this imply directly that all of the properties of $\tilde{f}$ translate to properties of $f$. This seems a little bit informal to me.

I also found that there is at least one missing citation in the line of upper bounds for the case of H&ouml;lder Smooth functions.

[1] Universal Gradient Methods for Stochastic Convex Optimization

**Questions:**

Please reference the weaknesses for my questions.

---

> ### Author Response · Authors · 2024-11-20
> **Reply to Official Review by Reviewer srAx**
>
> We thank the reviewer for the meticulous review and are incredibly grateful for the reviewer's effort in finding typos. We have fixed the typos and hope that they do not overshadow our core contributions: providing the tight lower bound for uniformly convex and high-order Hölder smooth functions. We sincerely hope that the reviewer can kindly consider raising the score if the following helps address some of the concerns.
>
> > The main issue for me that causes confusion in section 4 ... omitting $p$. I was wondering if there was a reason for this, as it is not discussed and in this setting
> $p \geq 1$, therefore for me it is not trivial to remove it.
>
> This is just a typo and we have added $p$ back throughout the derivation. We would kindly note that even though $p \geq 1$, it is nevertheless a constant and does not affect the order of our lower bound. That is, we indeed provide the tight lower bound in the $q > p + \nu$ setting.
>
> > Line 783, Line 230, Line 934, Line 293, Line 296, Line 317, Line 1011, 1019
>
> For lines 783, 230, 934, 293, 296, and 1011, we have fixed these typos. For line 1019, we have added the reference to Lemma 3 (ii). For line 317, we are indeed referring to Lemma 14 (i) in which the gradient of the smoothed function is explicitly defined.
>
> > For Section 5, I only have one question. Concerning lines 402,403, what do you mean by scaling of $\tilde{f}$ and how does this imply directly that all of the properties of $\tilde{f}$ translate to properties of $f$. This seems a little bit informal to me.
>
> By definition, $f$ is a scaling and rotation of $\tilde{f}$. Specifically, for $\mathbf{v}\_1, \cdots, \mathbf{v}\_{\tilde{T}}$, we can write for $V = [\mathbf{v}\_1, \cdots, \mathbf{v}\_{\tilde{T}}]$, $f(\mathbf{x}) = \frac{H}{2^{p+\nu+1} (p+\nu-1)!}\tilde{f}(V\mathbf{x})$, where $V$ is an orthonormal matrix. Therefore, the properties of $\tilde{f}$ translate to properties of $f$ with different constants. This is the same trick applied in [Arjevani et al. 2019], e.g., Eq. (10), Eq. (21).
>
> > Missing citation
>
> We thank the reviewer for pointing us to related papers. While we focus on the deterministic setting, we have also included this reference.

---

> > ### Comment · Reviewer_srAx · 2024-11-20
> >
> > I thank the authors for updating the manuscript and for their clarifications. I believe that this is an interesting work for the community and my only concern was the mistakes that could cause confusion to a potential reader. I have updated my score accordingly.
> >
> >    Given the clarification that the lower bounds apply to deterministic algorithms and the updates to the theorems statements in the draft I believe that the citation to [1] is not relevant anymore so feel free to omit it, if you prefer.

---

### Official Review · Reviewer_ApvT · 2024-10-27

**Soundness:** 3
**Presentation:** 4
**Contribution:** 2
**Rating:** 8
**Confidence:** 4

**Summary:**

The paper studies oracle complexity lower bounds for optimizing highly-smooth uniformly-convex functions with high-order methods.

Tight lower bounds are established in the "asymmetric" cases when $q\neq p+\nu$, where $q$ is the uniform-convexity degree, $\nu$ is the Holder-continuity degree, and $p$ is the oracle access degree.

**Strengths:**

This paper is well written. The results are motivated, well-explained and well-situated with respect to prior work.

I especially liked discussion such as the one appearing in lines 170-191 explaining the intuition in the lower bound construction, and the subtle challenges appearing with respect to prior work on similar subjects.

I haven't checked the math too carefully, but the authors make convincing points so that I believe the results should be solid.

**Weaknesses:**

One might argue that there isn't much technical novelty, as the two lower bound constructions are adaptations of previous know lower bounds (essentially the Nemirovski function and the Nesterov function), with relatively similar analyses to prior works.

Personally, I do not think this should prevent acceptance, as the authors explain what are the key differences, and the optimization community would benefit from these lower bounds being published.

**Questions:**

## Two inaccuracies
I recommend to slightly revise the paper according to the following points:

1) In line 168 it is argued that Gaussian smoothing is dim-free, whereas uniform (over an l2 ball) is not. I believe this might be wrong, since they are essentially the same. Gaussian smoothing is nearly the same as uniform smoothing over an l2 ball with radius $\approx\sqrt{d}$, since the distributions are nearly the same by concentration of measure. So it's just that the authors seem to be comparing a $\delta$-approximation smoothing to a $\delta\sqrt{d}$-approximation, which clearly wouldn't have the same smoothing parameters. Is this the case, or am I missing something? Similarly, I believe the same is true for sofrmax smoothing, which isn't really dim-free since the approximation parameter has a $\log(d)$ factor. So making is $\delta$-approximate would mean that the smoothness constant is $\approx\log(d)/\delta$, no? This is a milder logarithmic dependence as opposed to polynomial, but still not entirely dim-free.

2) As written, the proofs hold for deterministic algorithms. The authors do not discuss what is the algorithm class their lower bound holds for, and specifically I believe their results hold for deterministic (or more generally, zero-respecting) algorithms. The first lower bound based on the Nemirovski max function should hold also for randomized algorithm (with a quadratically larger dimension) based on so called "robust-zero-chain" arguments due to "Lower bounds for finding stationary points" by Carmon et al., but these are not given by the authors. Moreover, the second lower bound based on in the Nesterov function is not a robust zero-chain so it shouldn't hold beyond deterministic/zero-respecting algorithms, if I am not mistaken. The authors should discuss this in their paper.

## Additional question
- What seems to be preventing a lower bound for the symmetric $q=p+\nu$ case? Why do both proof techniques break? It would be great if the authors discuss this in the paper.


## Minor comments
- Notation: $\|\cdot\|$ refers to operator $\ell_2$ norms, maybe should emphasize this (just adding the word "operator") to avoid confusion with Frobenius l2.
- Lemma 1 (i) - the authors probably forgot to assume the function is $L$-Lipschitz.

---

> ### Author Response · Authors · 2024-11-20
> **Reply to Official Review by Reviewer ApvT [1/2]**
>
> We thank the reviewers for acknowledging our contributions. Here we provide further clarifications to address the reviewer's questions and concerns.
>
> > One might argue that there isn't much technical novelty, as the two lower bound constructions are adaptations of previous know lower bounds (essentially the Nemirovski function and the Nesterov function), with relatively similar analyses to prior works. Personally, I do not think this should prevent acceptance, as the authors explain what are the key differences, and the optimization community would benefit from these lower bounds being published.
>
> We appreciate the reviewer for acknowledging our differences and novelties, as we believe that the Nemirovski function and the Nesterov function serve more as general frameworks for proving lower bounds, and numerous papers on lower bounds in various settings (e.g., deterministic and stochastic, first-order and high-order, etc.) build on these constructions. It is the specific techniques applied along the way that allow for achieving tight results under different settings, e.g., in our $q>p+\nu$ case, the application of the newly proposed truncated Gaussian smoothing, the choice of $\ell\_\infty$ truncation, and proving its smoothing properties in Lemma 1, which we believe is technically non-trivial.
>
> > In line 168 it is argued that Gaussian smoothing is dim-free, whereas uniform (over an l2 ball) is not. I believe this might be wrong, since they are essentially the same. Gaussian smoothing is nearly the same as uniform smoothing over an l2 ball with radius
> $\sqrt{d}$, since the distributions are nearly the same by concentration of measure. So it's just that the authors seem to be comparing a $\delta$-approximation smoothing to a $\delta\sqrt{d}$-approximation, which clearly wouldn't have the same smoothing parameters. Is this the case, or am I missing something? Similarly, I believe the same is true for softmax smoothing, which isn't really dim-free since the approximation parameter has a $\log d$
>  factor. So making is $\delta$-approximate would mean that the smoothness constant is $\log d / \delta$, no? This is a milder logarithmic dependence as opposed to polynomial, but still not entirely dim-free.
>
>
> We would kindly note that we are not saying uniform smoothing over **any** $\ell\_2$-ball is dimension-dependent, but over a **unit** $\ell\_2$-ball (in lines 161 to 163), that is, a $\ell\_2$-ball with radius 1, in contrast to the radius $\sqrt{d}$ the reviewer proposed. Indeed, uniform smoothing over an $\ell\_2$-ball with radius $\sqrt{d}$ is also dimension-free by setting $u=\sqrt{d}$ in Lemma 8 of [Duchi et. al. 2012], which justifies why Gaussian smoothing is also dimension-free. And by "dimension-free", we mean exactly that there is not a $d$ factor in the smoothness constants in Lemma 9 of [Duchi et. al. 2012] with $u$ not dependent on $d$ or Theorem 7 of [Bullins 2020] with $\mu$ not dependent on $d$. Furthermore, the $\delta$ in our paper measures the radius of the local region, i.e., $\Vert\mathbf{x} - \mathbf{x}_s\Vert \leq \frac{\delta}{2}$ in lines 300-301, not the approximation error of $g(\mathbf{x}) - G(\mathbf{x})$ in lines 233-234 Lemma 3 (ii).
>
> Nevertheless, we agree that there is a connection between the approximation and the smoothness constant, where one can move the $d$ in the smoothness constant to approximation error by controlling the degree of smoothing, e.g., as the reviewer pointed out, for softmax smoothing, by setting the $\mu = \frac{1}{\log d}$ in Theorem 7 of [Bullins 2020], the approximation error $f(\mathbf{x}) \leq f^{softmax}(\mathbf{x}) \leq f(\mathbf{x}) + p\mu \log d$ reduces to a constant $p \mu \log d = p$ whereas the smoothness constant becomes $d$-dependent. With that said, we regard the parameter controlling the degree of smoothing as a constant parameter to begin with in this paper.
>
> In addition, the question raised by the reviewer has led the authors to further examine whether uniform smoothing within an $\ell\_2$-ball with radius $\sqrt{d}$ would work, e.g., Lemma 8 of [Duchi et. al. 2012] is identical to our Lemma 14 if $u = \sqrt{d}$. The answer is negative because the radius of the local region would then be $d$-dependent, i.e., $\Vert \mathbf{x} - \mathbf{x}_s \Vert\_2 \leq 2\sqrt{d}$ in lines 300-301. This is where the $\delta$ in lines 300-301 becomes $d$-dependent, (instead of the approximation error the reviewer referred to) and breaks the derivation there.

---

> ### Author Response · Authors · 2024-11-20
> **Reply to Official Review by Reviewer ApvT [2/2]**
>
> > As written, the proofs hold for deterministic algorithms. The authors do not discuss what is the algorithm class their lower bound holds for, and specifically I believe their results hold for deterministic (or more generally, zero-respecting) algorithms.
>
> We thank the reviewer for pointing this out. Indeed we consider deterministic algorithms in this paper. We have now specified the algorithm class in Theorem 1 and 2. We also thank the reviewer for bringing up the adaptability to randomized algorithms. We will add relevant discussion in the revised version, and omit for now to keep the line number consistent.
>
>
> > What seems to be preventing a lower bound for the symmetric $q = p+\nu$ case? Why do both proof techniques break? It would be great if the authors discuss this in the paper.
>
> We thank the reviewer for expressing interest in our future work. Here we briefly elaborate on the challenges lying ahead in this direction. In the $q=p+\nu$ case, one would expect a linear convergence rate of $\Omega\left(\left(\frac{H}{\sigma}\right)^\frac{2}{3(p+\nu)-2} \log \left(\frac{1}{\epsilon}\right)\right)$ according to the high-order upper bound in [Song et. al., 2022], as well as the first-order lower bound $\Omega\left(\left(\frac{H}{\sigma}\right)^\frac{1}{2} \log \left(\frac{1}{\epsilon}\right)\right)$ [Nesterov, 2018, Section 2.1.4.] when $p=1$ and $\nu=1$. The asymmetric cases do not naturally yield a linear rate when $q=p+\nu$. On top of achieving linear convergence, one would also expect a tight dependence on the condition number, e.g., $\sqrt{\frac{H}{\sigma}}$ in the first-order setting. Such a result, i.e., the factor $\sqrt{\frac{H}{\sigma}}$, is derived from solving a quadratic equation with the closed-form quadratic formula, as demonstrated in Eq. (2.1.33) Section 2.1.4. in [Nesterov, 2018]. Yet in high-order scenarios, the natural extension would involve solving a high-ordered equation, for which there is no closed-form formula. As a result, we are in the process of searching for alternative ways to derive the linear convergence lower bound for the $q=p+\nu$ case with a tight dependence on the condition number.
>
> > Minor comments
>
> We have added the word "operator", and the $L$-Lipschitzness is assumed at the beginning of Lemma 1.

---

> > ### Comment · Reviewer_ApvT · 2024-11-26
> > **Response to rebuttal**
> >
> > I thank the authors for their detailed response. I maintain my score.

---

### Official Review · Reviewer_JCi9 · 2024-11-04

**Soundness:** 4
**Presentation:** 4
**Contribution:** 4
**Rating:** 8
**Confidence:** 3

**Summary:**

The paper studies oracle lower bounds for minimizing high-order holder smooth and uniform convex objective functions using pth-order oracles. The lower bounds are proven by explicit construction of hard problem instances, and match known upper bounds for the function classes considered.

**Strengths:**

The paper presents the problem setup, ideas, and results clearly. The lower bounds obtained are tight, and contribute to the oracle complexity of convex optimization literature.

**Weaknesses:**

The paper provides oracle lower bounds for function classes not studied before (in terms of lower bound). The construction of the hard instances, although novel (e.g. using the truncated Gaussian for the smooth), inherits some ideas used previously. For instance, the function in line 218 - line 220 is also used in [1]. It would be great if the authors can provide a brief summary/comparison of which parts of the lower bound construction are used before, and which parts are new.


[1] Nikita Doikov. Lower complexity bounds for minimizing regularized functions. arXiv preprint arXiv:2202.04545, 2022.

**Questions:**

Is the norm in $||\nabla^i f||$ (in line 201 and many other places) the operator norm?

---

> ### Author Response · Authors · 2024-11-20
> **Reply to the Official Review by Reviewer JCi9**
>
> We thank the reviewer for acknowledging our contributions and providing constructive comments. To further address the reviewer's comments, we would like to clarify and justify the differences between our work and previous works.
>
> > The construction of the hard instances, although novel (e.g. using the truncated Gaussian for the smooth), inherits some ideas used previously. For instance, the function in line 218 - line 220 is also used in [1].
>
> While the reviewer has rightfully pointed out that [1] follows similar initial steps to construct a hard function, we would kindly note that such construction serves more as a general framework for proving lower bounds, and is adopted not only in [1] but also in [2-4] for results under a wide variety of problem settings, and even dates back to earlier works [5]. Though this framework is quite important in general for proving lower bounds, it is the specific techniques applied along the way of construction that allow for achieving tight results under different settings, e.g., in our $q>p+\nu$ case, the application of the newly proposed truncated Gaussian smoothing. The choice of $\ell\_\infty$ truncation as well as proving its smoothing properties in Lemma 1, as we believe, is fairly mathematically involved and technically non-trivial. We have briefly summarized the above in lines 137-143, and are happy to further highlight our novelties in the revised version.
>
> > Questions: Is the norm in
>  (in line 201 and many other places) the operator norm?
>
> Yes. We have further specified in Section 3 line 110 that $\Vert \cdot \Vert$ denotes $\ell\_2$ operator norm.
>
> ----
>
> References:
>
> [1] Nikita Doikov. Lower complexity bounds for minimizing regularized functions. arXiv preprint arXiv:2202.04545, 2022.
>
> [2] Crist ´obal Guzm ´an and Arkadi Nemirovski. On lower complexity bounds for large-scale smooth
> convex optimization. Journal of Complexity, 31(1):1–14, 2015.
>
> [3] Naman Agarwal and Elad Hazan. Lower bounds for higher-order convex optimization. In Conference
> On Learning Theory, pp. 774–792. PMLR, 2018.
>
> [4] Ankit Garg, Robin Kothari, Praneeth Netrapalli, and Suhail Sherif. Near-optimal lower bounds for
> convex optimization for all orders of smoothness. Advances in Neural Information Processing
> Systems, 34:29874–29884, 2021.
>
> [5] Arkaddii S Nemirovskii and Yu E Nesterov. Optimal methods of smooth convex minimization. USSR
> Computational Mathematics and Mathematical Physics, 25(2):21–30, 1985.

---

> > ### Comment · Reviewer_JCi9 · 2024-11-25
> >
> > Thank the authors for the comments! I'll keep my score.

---

### Official Review · Reviewer_JCuX · 2024-11-10

**Soundness:** 3
**Presentation:** 3
**Contribution:** 4
**Rating:** 8
**Confidence:** 4

**Summary:**

This paper investigates the oracle complexity of minimizing high-order Hölder smooth and uniformly convex functions, particularly under two asymmetric cases: when the degree of uniform convexity $q$ is greater than $p + \nu$, and when $q < p + \nu$, where $p$ is the order of smoothness, and $\nu$ is the Hölder degree. The authors establish tight lower bounds for these cases, generalizing previous lower bounds for uniformly convex functions with first- and second-order smoothness. They employ the $\ell_\infty$-ball-truncated Gaussian smoothing operator to achieve dimension-free smoothness for their hard function construction, providing theoretical insights into high-order optimization.

**Strengths:**

The problem studied in the paper is both important and well-motivated. The paper provides a rigorous analysis with novel contributions to the study of oracle complexity for asymmetric high-order convex functions, extending previous research by offering a general framework. The authors carefully construct proofs to demonstrate their lower bounds, presenting the results in a way that is accessible for further theoretical work. Additionally, the use of truncated Gaussian smoothing introduces an innovative approach to managing complexity in high-order smoothness contexts, achieving dimension-free smoothness without dependence on dimensionality.

**Weaknesses:**

The paper does not cover the case of $q= p + \nu$

**Questions:**

How might future work address the remaining case of $q = p + \nu$, and what specific challenges are anticipated?

---

> ### Author Response · Authors · 2024-11-20
> **Reply to the Official Review by Reviewer JCuX**
>
> We thank the reviewer for acknowledging our contributions and expressing interest in our future work on the $q=p+\nu$ case. Here we briefly elaborate on the challenges lying ahead in this direction. In the $q=p+\nu$ case, one would expect a linear convergence rate of $\Omega\left(\left(\frac{H}{\sigma}\right)^\frac{2}{3(p+\nu)-2} \log \left(\frac{1}{\epsilon}\right)\right)$ according to the high-order upper bound in [Song et. al., 2021], as well as the first-order lower bound $\Omega\left(\left(\frac{H}{\sigma}\right)^\frac{1}{2} \log \left(\frac{1}{\epsilon}\right)\right)$ [Nesterov, 2018, Section 2.1.4.] when $p=1$ and $\nu=1$. On top of achieving linear convergence, one would also expect a tight dependence on the condition number, e.g., $\sqrt{\frac{H}{\sigma}}$ in the first-order setting. Such a result, i.e., the factor $\sqrt{\frac{H}{\sigma}}$, is derived from solving a quadratic equation with the closed-form quadratic formula, as demonstrated in Eq. (2.1.33) Section 2.1.4. in [Nesterov, 2018]. Yet in high-order scenarios, the natural extension would involve solving a high-ordered equation, for which there is no closed-form formula. As a result, we are in the process of searching for alternative ways to derive the linear convergence lower bound for the $q=p+\nu$ case with a tight dependence on the condition number. We will incorporate the discussion in the revised version and omit it for now to keep the line number consistent.
>
> ----
>
> References:
>
> Song et. al. Unified acceleration of high-order algorithms under general holder continuity. SIAM Journal on Optimization, 2022
>
> Nesterov, Yurii. Lectures on convex optimization. 2018.

---

> > ### Comment · Reviewer_JCuX · 2024-11-26
> >
> > I thank the authors for the response. I maintain my score.

---

### Meta-Review · Area_Chair_T5ng · 2024-12-21

**Metareview:**

This is a theoretical paper that studies the oracle complexity of minimizing convex functions with varying degrees of high-order smoothness, p, and degrees of uniform convexity, q. The oracle has access to pth order derivatives. While the authors lower bounds do not cover all ranges of parameters, they are tight in the ranges covered, and thus the reviewers agreed that the paper makes fundamental progress on an important challenge in convex optimization. Moreover, it was agreed that the paper was well-written, with intuition provided throughout, and the problems well motivated. We would be happy to have this paper appear at ICLR.

**Additional Comments On Reviewer Discussion:**

The authors clarified minor questions and some typos. The reviewers felt positively about the paper before discussion.

---

### Decision · Program_Chairs · 2025-01-22

Accept (Oral)